# LoRANN: Low-Rank Matrix Factorization for Approximate Nearest Neighbor Search

**Elias Jääsaari** [†]   **Ville Hyvönen** [‡]   **Teemu Roos** [†]

[†]Department of Computer Science, University of Helsinki
[‡]Department of Computer Science, Aalto University
`{elias.jaasaari,teemu.roos}@helsinki.fi; ville.2.hyvonen@aalto.fi`

## Abstract

Approximate nearest neighbor (ANN) search is a key component in many modern machine learning pipelines; recent use cases include retrieval-augmented generation (RAG) and vector databases. Clustering-based ANN algorithms, that use score computation methods based on product quantization (PQ), are often used in industrial-scale applications due to their scalability and suitability for distributed and disk-based implementations. However, they have slower query times than the leading graph-based ANN algorithms. In this work, we propose a new supervised score computation method based on the observation that inner product approximation is a multivariate (multi-output) regression problem that can be solved efficiently by reduced-rank regression. Our experiments show that on modern high-dimensional data sets, the proposed reduced-rank regression (RRR) method is superior to PQ in both query latency and memory usage. We also introduce `LoRANN`[1], a clustering-based ANN library that leverages the proposed score computation method. `LoRANN` is competitive with the leading graph-based algorithms and outperforms the state-of-the-art GPU ANN methods on high-dimensional data sets.

## 1 Introduction

In modern machine learning applications, data is often stored as embeddings, i.e., as vectors in a high-dimensional vector space where representations of semantically similar items are close to each other. Consequently, similarity search in high-dimensional vector spaces is a key algorithmic primitive used in many pipelines, such as semantic search engines and recommendation systems. Since the data sets are usually both large and high-dimensional, *approximate* nearest neighbor (ANN) search is deployed to speed up similarity search in many applications (Li et al., 2019).

Recent use cases of ANN search include retrieval-augmented generation (RAG) (Lewis et al., 2020; Guu et al., 2020; Borgeaud et al., 2022; Shi et al., 2024) and approximate attention computation in Transformer-based architectures (Kitaev et al., 2020; Vyas et al., 2020; Roy et al., 2021). ANN search is also a key operation in vector databases that are used to store embeddings for industrial-scale applications (see, e.g., Wang et al., 2021; Guo et al., 2022; Pan et al., 2024).

The state-of-the-art methods for ANN search can be classified into clustering-based and graph-based algorithms (for a recent survey, see Bruch, 2024). In the comprehensive ANN benchmark (Aumüller et al., 2020), the leading graph algorithms HNSW (Malkov and Yashunin, 2018) and NGT (Iwasaki and Miyazaki, 2018) have faster query times than clustering-based algorithms. However, clustering-based algorithms are often used in industrial-scale applications (see, e.g., Chen et al., 2021; Douze et al., 2024) due to their smaller memory footprints and faster index construction times. They are

---

[1]`https://github.com/ejaasaari/lorann`

also suitable for distributed implementations and hybrid solutions that use persistent storage such as SSDs or blob storage (Chen et al., 2021; Gottesbüren et al., 2024).

The key components of clustering-based algorithms are *clustering* and *score computation*. In the indexing phase, the corpus, i.e., the data set from which the nearest neighbors are searched, is partitioned via clustering. In the query phase, $w$ clusters are selected and the corpus points that belong to the selected clusters are scored. The points with the highest scores are selected into a candidate set that can be further re-ranked. The state-of-the-art clustering-based algorithms (Jegou et al., 2011; Guo et al., 2020; Sun et al., 2023) use variants of *product quantization* (PQ) (Jegou et al., 2011) with highly optimized implementations (see, e.g., André et al., 2015) for score computation.

In this article, we propose a supervised score computation method that improves the query latency of clustering-based ANN methods, making them competitive with the leading graph algorithms. Our key observation is that estimating the dissimilarities between the query point and the cluster points is a multivariate (multi-output) regression problem. For the most common dissimilarity measures, it is sufficient to estimate the inner products between the query point and the cluster points, and the ordinary least squares (OLS) estimate is the exact solution for this regression problem. The proposed method approximates the OLS solution by reduced-rank regression (Izenman, 1975). We further approximate the reduced-rank regression estimates using 8-bit integer computations, improving query latency and memory consumption. Reduced-rank regression (RRR) is a simpler method than PQ, and our experimental results show that it is faster at any given recall level and memory usage.

To make our work available for practical applications of ANN search, we introduce `LoRANN`, a clustering-based ANN library that leverages the proposed score computation method. Since the low memory usage and the simple computational structure of reduced-rank regression make it well-suited for GPUs, `LoRANN` also includes a GPU implementation of the proposed method.

In summary, our contributions are:

- We propose reduced-rank regression (RRR) as a new supervised score computation method for clustering-based ANN search (see Section 3).
- We verify experimentally that RRR outperforms PQ both at the optimal hyperparameters (Section 7.1.1) and at fixed memory consumption (Section 7.1.2), and that it naturally adapts to the query distribution in the out-of-distribution (OOD) setting (Section 5).
- We introduce `LoRANN`, a clustering-based ANN library that contains efficient CPU and GPU implementations of the proposed score computation method (Section 4).
- We show that `LoRANN` outperforms the leading clustering-based libraries Faiss (Douze et al., 2024) and ScaNN (Guo et al., 2020), and has faster query times than the leading graph-based library GLASS at recall levels under 90% on most data sets (Section 7.2.2). `LoRANN` outperforms the SOTA GPU methods on high-dimensional data (Section 7.2.3).

## 2 Background

In this section, we first review the notation of approximate nearest neighbor (ANN) search and then describe the standard structure of clustering-based ANN algorithms.

### 2.1 ANN search

Let $\mathbf{x} \in \mathbb{R}^d$ be a query point, and let $\{\mathbf{c}_j\}_{j=1}^m \subset \mathbb{R}^d$ be the *corpus*, i.e., the set of points from which the nearest neighbors are retrieved. The $k$ nearest neighbors of $\mathbf{x}$ are defined as

$$\text{NN}_k(\mathbf{x}; \{\mathbf{c}_j\}_{j=1}^m, d) := \{j \in [m] \,:\, d(\mathbf{x}, \mathbf{c}_j) \leq d(\mathbf{x}, \mathbf{c}_{(k)})\}, \tag{1}$$

where $d : \mathbb{R}^d \to \mathbb{R}$ is a dissimilarity measure, $\mathbf{c}_{(1)}, \ldots, \mathbf{c}_{(m)}$ denote the corpus points that are ordered in ascending order w.r.t. their dissimilarity to the query point $\mathbf{x}$, and $[m] := \{1, \ldots, m\}$.

Commonly used dissimilarity measures are the Euclidean distance $d(\mathbf{a}, \mathbf{b}) = \|\mathbf{a} - \mathbf{b}\|_2$, the (negative) inner product $d(\mathbf{a}, \mathbf{b}) = -\langle \mathbf{a}, \mathbf{b} \rangle$, and the cosine (angular) distance $d(\mathbf{a}, \mathbf{b}) = 1 - \langle \mathbf{a}/\|\mathbf{a}\|_2, \mathbf{b}/\|\mathbf{b}\|_2 \rangle$. The special case where the dissimilarity measure is the negative inner product is often called maximum inner product search (MIPS) (see, e.g., Guo et al., 2020; Lu et al., 2023; Zhao et al., 2023). In ANN search, the exact solution $\text{NN}_k(\mathbf{x}; \{\mathbf{c}_j\}_{j=1}^m, d)$ is approximated.

The effectiveness of ANN algorithms is typically measured by *recall*, i.e., the fraction of the true $k$ nearest neighbors returned by the algorithm. The efficiency is measured by query latency or, equivalently, by queries per second (QPS) (see, e.g., Li et al., 2019; Aumüller et al., 2020).

## 2.2 Clustering-based ANN search

Partitioning-based ANN algorithms, such as tree-based (e.g., Muja and Lowe, 2014; Dasgupta and Sinha, 2015; Jääsaari et al., 2019) and hashing-based algorithms (e.g., Charikar, 2002; Weiss et al., 2008; Liu et al., 2011), build an index by partitioning the corpus into $L$ elements. In the query phase, they use a routing function $\tau : \mathbb{R}^d \to [L]^w$ to assign the query point into $w$ partition elements.

Clustering-based ANN algorithms (see, e.g., Bruch, 2024, Chapter 7) are partitioning-based ANN algorithms that partition the corpus via clustering. ANN indexes based on clustering are also often called *inverted file* (IVF) indexes (Jegou et al., 2011). The most commonly used clustering method is $k$-means clustering, specifically standard $k$-means when the dissimilarity measure $d$ is the Euclidean distance, and spherical $k$-means (Dhillon and Modha, 2001) when $d$ is the inner product or the cosine distance. While there exists recent exploratory work on alternative query routing methods (Gottesbüren et al., 2024; Vecchiato et al., 2024; Bruch et al., 2024), the most common method is centroid-based routing where, for the set $\{\boldsymbol{\mu}_l\}_{l=1}^L$ of cluster centroids,

$$\tau(\mathbf{x}) = \text{NN}_w(\mathbf{x}; \{\boldsymbol{\mu}_l\}_{l=1}^L, d),$$

i.e., the $w$ clusters whose centroids are closest to the query point are selected. We follow this standard practice by using $k$-means clustering with centroid-based routing. However, the proposed score computation method can be combined with any partitioning and routing method.

After routing, the corpus points that belong to the selected $w$ clusters are scored (see Section 6 for a discussion of score computation methods), and the $t$ highest scoring points are selected into the candidate set. This candidate set can be re-ranked by evaluating the true dissimilarities between $\mathbf{x}$ and the candidate set points. Finally, the $k$ most similar points are returned as the approximate $k$-nn.

# 3 Reduced-rank regression

In this section, we derive the proposed supervised score computation method. First, we formulate dissimilarity approximation as a multivariate regression problem. We then show how the exact OLS solution to this problem can be approximated by reduced-rank regression (RRR). Finally, we show how RRR can be implemented efficiently using 8-bit integer vector-matrix multiplications.

## 3.1 Dissimilarity approximation as a multivariate regression problem

We consider the task of approximating the dissimilarities $d(\mathbf{x}, \mathbf{c}_j)$ between the query point $\mathbf{x}$ and the corpus points $\mathbf{c}_j$ that belong to the $l$th cluster. Denote the set of the indices of these corpus points by $I_l$, their number by $m_l := |I_l|$, and the matrix containing them as rows by $\mathbf{C}_l \in \mathbb{R}^{m_l \times d}$. In what follows, to avoid cluttering the notation, we drop the subscript $l$ denoting the cluster from matrices, e.g., we denote $\mathbf{C}_l$ by $\mathbf{C}$. We also assume, w.l.o.g., that the corpus is indexed so that $I_l = \{1, \ldots, m_l\}$. This task can now be formulated as a multivariate regression problem where the output is defined as a $1 \times m_l$ matrix $\mathbf{y} = [y_1 \ldots y_{m_l}]$, where $y_j = d(\mathbf{x}, \mathbf{c}_j)$ for each $j = 1, \ldots, m_l$.

We consider the cases where $d$ is the (negative) inner product, the Euclidean distance, or the cosine (angular) distance. In all three cases, it is sufficient to estimate the inner products. For Euclidean distance, $\arg\min_{j \in I_l} \|\mathbf{x} - \mathbf{c}_j\|_2 = \arg\min_{j \in I_l}(-2\mathbf{x}^T\mathbf{c}_j + \|\mathbf{c}_j\|_2^2)$, where the norms $\|\mathbf{c}_j\|_2$ can be precomputed. For cosine distance, $\arg\min_{j \in I_l}(1 - \cos(\mathbf{x}, \mathbf{c}_j)) = \arg\min_{j \in I_l}(-\mathbf{x}^T\mathbf{c}_j)$ if the corpus points are normalized to have unit norm.

## 3.2 Reduced-rank regression solution

We approximate the exact solution $\mathbf{y} = \mathbf{x}^T\mathbf{C}^T$ of the regression problem defined in the previous section by a low-rank approximation. We assume the standard supervised learning setting, i.e., that we have a sample $\{\mathbf{x}_i\}_{i=1}^n$ from the query distribution $\mathcal{Q}$ (the corpus can also be used as the training set if no separate training set is available). To train the $l$th model, we use all the training set points that

are routed into the $l$th cluster. When the standard centroid-based routing is used, these are the training set points that have $\boldsymbol{\mu}_l$, i.e., the centroid of the $l$th cluster, among their $w$ closest centroids. Denote the set of indices of these training set points by $J_l := \{i \in [n] : l \in \tau(\mathbf{x}_i)\}$, and their number by $n_l := |J_l|$. The output values of the training set of the $l$th model are given by $\mathbf{Y} := \mathbf{X}\mathbf{C}^T \in \mathbb{R}^{n_l \times m_l}$, where we denote by $\mathbf{X} \in \mathbb{R}^{n_l \times d}$ the matrix containing the training set points $\{\mathbf{x}_i\}_{i \in J_l}$ as rows.

To approximate the dissimilarities between the query point $\mathbf{x}$ and the cluster points $\{\mathbf{c}_j\}_{j \in I_l}$, we consider the linear model $\mathbf{x}^T\boldsymbol{\beta}$, where $\boldsymbol{\beta} \in \mathbb{R}^{d \times m_l}$ is a matrix containing the parameters of the model, and minimize the mean squared error $\mathbb{E}_{\mathbf{x} \sim \mathcal{Q}}[\|\mathbf{y} - \mathbf{x}^T\boldsymbol{\beta}\|_2^2 \, \mathbb{I}\{l \in \tau(\mathbf{x})\}]$ (the indicator function selects the query points that routed into the $l$th cluster). The unconstrained least squares solution $\hat{\boldsymbol{\beta}}_{\text{OLS}} = \mathbf{C}^T$ reproduces the exact inner products $\mathbf{y} = \mathbf{x}^T\mathbf{C}^T$. In order to reduce the computational complexity of evaluating the model predictions, we constrain the rank of the parameter matrix: $\text{rank}(\boldsymbol{\beta}) \leq r < \min(d, m_l)$. Under this constraint, the parameter matrix can be written using a low-rank matrix factorization $\boldsymbol{\beta} = \mathbf{A}\mathbf{B}$, where $\mathbf{A} \in \mathbb{R}^{d \times r}$ and $\mathbf{B} \in \mathbb{R}^{r \times m_l}$, and, consequently, the model predictions $\hat{\mathbf{y}} = (\mathbf{x}^T\mathbf{A})\mathbf{B}$ can be computed with $\Theta(r(d + m_l))$ operations. When the rank $r$ is sufficiently low, this is significantly faster than computing the exact inner products which requires $\Theta(dm_l)$ operations. Our experiments (Section 7) indicate that fixing this hyperparameter to $r = 32$ works well with a wide range of data sets encompassing dimensionalities between 128 and 1536.

The optimal low-rank solution can be found by minimizing the training loss

$$\hat{\boldsymbol{\beta}}_{\text{RRR}} = \underset{\boldsymbol{\beta} \, : \, \text{rank}(\boldsymbol{\beta}) \leq r}{\arg\min} \|\mathbf{Y} - \mathbf{X}\boldsymbol{\beta}\|_F^2,$$

where $\|\cdot\|_F$ is the Frobenius norm. This is the well-known *reduced-rank regression* problem (Izenman, 1975). Denote the singular value decomposition (SVD) of $\mathbf{Y}$ as $\mathbf{Y} = \mathbf{U}\boldsymbol{\Sigma}\mathbf{V}^T$, where $\boldsymbol{\Sigma}$ is a non-negative diagonal matrix and $\mathbf{U}$ and $\mathbf{V}$ are orthonormal matrices. The standard reduced-rank regression solution is $\hat{\boldsymbol{\beta}}_{\text{RRR}} = \hat{\boldsymbol{\beta}}_{\text{OLS}}\mathbf{V}_r\mathbf{V}_r^T = \mathbf{C}^T\mathbf{V}_r\mathbf{V}_r^T = \mathbf{A}\mathbf{B}$, where $\mathbf{V}_r \in \mathbb{R}^{m_l \times r}$ denotes the matrix that contains the first $r$ columns of $\mathbf{V}$ (i.e., the first $r$ right singular vectors of the least squares fit $\mathbf{Y} = \mathbf{X}\mathbf{C}^T$), $\mathbf{A} := \mathbf{C}^T\mathbf{V}_r$, and $\mathbf{B} := \mathbf{V}_r^T$. In practice, we use a fast randomized algorithm (Halko et al., 2011) to compute only $\mathbf{V}_r$ instead of the full SVD. Observe that the reduced-rank regression solution is different from the most obvious low-rank matrix factorization of the OLS solution computed via an SVD of the matrix $\mathbf{C}$ (see Appendix A).

### 3.3   8-bit quantization

The simple computational structure of the reduced-rank regression solution enables us to further improve its query latency and memory consumption by using integer quantization. For each cluster, we quantize the matrices $\mathbf{A}$ and $\mathbf{B}$ to 8-bit integer precision. By also quantizing the query vector $\mathbf{x}$, the model prediction $\hat{\mathbf{y}} = \mathbf{x}^T\hat{\boldsymbol{\beta}}_{\text{RRR}} = (\mathbf{x}^T\mathbf{A})\mathbf{B}$ can be computed efficiently in two 8-bit integer vector-matrix products. We use *absmax quantization* (e.g., Dettmers et al., 2022), where the elements of a vector $\mathbf{x}$ are scaled to the range $[-127, 127]$ by multiplying with a constant $c_{\mathbf{x}}$ such that $\mathbf{x}_{i8} = \lfloor (127/\|\mathbf{x}_{f32}\|_\infty) \cdot \mathbf{x}_{f32} \rceil = \lfloor c_{\mathbf{x}}\mathbf{x}_{f32} \rceil$, where $\lfloor\cdot\rceil$ denotes rounding to the nearest integer.

We quantize the matrices $\mathbf{A}$ and $\mathbf{B}$ by applying absmax quantization to each column of the given matrix, resulting in vectors $\mathbf{c}_{\mathbf{A}}$ and $\mathbf{c}_{\mathbf{B}}$ of scaling constants. We can then recover a 32-bit floating-point approximation to the vector-matrix product $\mathbf{r} = \mathbf{x}^T\mathbf{A}$ with

$$\mathbf{r}_{f32} \approx \frac{1}{c_{\mathbf{x}}}\mathbf{r}_{i32} \oslash \mathbf{c}_{\mathbf{A}} =: \mathbf{s} \odot \mathbf{r}_{i32} = \mathbf{s} \odot \mathbf{x}_{i8}^T\mathbf{A}_{i8} = \mathbf{s} \odot Q(\mathbf{x})^T Q(\mathbf{A}_{f32}),$$

where $Q(\cdot)$ denotes absmax quantization, and $\oslash$ and $\odot$ denote element-wise division and multiplication, respectively. To compute $\hat{\mathbf{y}} = \mathbf{r}^T\mathbf{B}$ in the same fashion, we can first re-quantize $\mathbf{r}$.

To ensure minimal loss of precision from the quantization, we rotate $\mathbf{x}$ before quantization by multiplying $\mathbf{x}$ with a random rotation matrix; this spreads the variance among the dimensions of $\mathbf{x}$. Similarly, we rotate the vector $\mathbf{r}$ resulting from the first product $\mathbf{r} = \mathbf{x}^T\mathbf{A}$ before re-quantization. Since $\mathbf{A} = \mathbf{C}^T\mathbf{V}_r$ and $\mathbf{B} = \mathbf{V}_r^T$, we can rotate $\mathbf{r}$ by rotating $\mathbf{V}_r$ beforehand at no extra cost.

**Memory usage**   Storing each matrix $\mathbf{A}_{i8} \in [\mathbb{Z}]_{256}^{d \times r}$ takes $dr$ bytes and each matrix $\mathbf{B}_{i8} \in [\mathbb{Z}]_{256}^{r \times m_l}$ takes $rm_l$ bytes. Thus in a clustering-based ANN index with $L$ clusters and $m$ corpus points, the total memory consumption of RRR is of order $Ldr + rm$ bytes. In our experiments (Section 7), we use $r = 32$ for all data sets, while $L$ is typically of order $\sqrt{m}$.

# 4   LoRANN

In this section, we describe the additional implementation details of `LoRANN`, an open-source library that combines the standard template of clustering-based ANN search described in Section 2.2 with the score computation method described in Section 3. Using dimensionality reduction is particularly efficient for RRR (Section 4.1) and works well with 8-bit integer quantization (Section 4.2). Finally, we describe the GPU implementation of `LoRANN` (Section 4.3).

## 4.1   Dimensionality reduction

With a moderate-sized corpus and high-dimensional data, computing the first vector-matrix product of the model prediction $\hat{\mathbf{y}} = (\mathbf{x}^T\mathbf{A})\mathbf{B}$ can be more expensive than the second. In `LoRANN`, we further approximate the product by first projecting the query into a lower-dimensional space. We use the projection matrix $\mathbf{W}_s \in \mathbb{R}^{d \times s}$ whose columns are the first $s$ eigenvectors of $\mathbf{X}_{\text{global}}^T\mathbf{X}_{\text{global}}$, where $\mathbf{X}_{\text{global}} \in \mathbb{R}^{n \times d}$ is the matrix containing the training set points $\{\mathbf{x}_i\}_{i=1}^n$. To estimate the reduced-rank regression models, we use the $s$-dimensional approximations $\tilde{\mathbf{x}}_i = \mathbf{W}_s^T\mathbf{x}_i$ as inputs, but the true inner products $\mathbf{x}_i^T\mathbf{c}_j$ as outputs. In this case, the reduced-rank regression estimate of the $l$th model is $\hat{\boldsymbol{\beta}}_{\text{RRR}} = \hat{\boldsymbol{\beta}}_{\text{OLS}}\mathbf{V}_r\mathbf{V}_r^T = (\mathbf{X}\mathbf{W}_s)^\dagger\mathbf{Y}\mathbf{V}_r\mathbf{V}_r^T \in \mathbb{R}^{s \times m_l}$, where $\hat{\boldsymbol{\beta}}_{\text{OLS}} := (\mathbf{X}\mathbf{W}_s)^\dagger\mathbf{Y}$ is the full-rank solution, and $\mathbf{V}_r \in \mathbb{R}^{m_l \times r}$ is the matrix whose columns are the first $r$ right singular vectors of $\mathbf{Y}$. Thus, now $\mathbf{A} := (\mathbf{X}\mathbf{W}_s)^\dagger\mathbf{Y}\mathbf{V}_r \in \mathbb{R}^{s \times r}$ and $\mathbf{B} := \mathbf{V}_r^T \in \mathbb{R}^{r \times m_l}$.

We observe that computing query-to-centroid distances in the $s$-dimensional space yields a minor performance improvement. In the indexing phase, we perform $k$-means clustering using $\tilde{\mathbf{c}}_j = \mathbf{W}_s^T\mathbf{c}_j$. In the query phase, the $s$-dimensional approximation of the query point, $\tilde{\mathbf{x}} = \mathbf{W}_s^T\mathbf{x}$, is used to compute the distances to the cluster centroids and the predictions $\hat{\mathbf{y}} = \tilde{\mathbf{x}}^T\hat{\boldsymbol{\beta}}_{\text{RRR}}$. The original $d$-dimensional query point $\mathbf{x}$ is used for the dissimilarity evaluations in the final re-ranking step.

## 4.2   Quantization implementation

The dimensionality reduction works particularly well with the 8-bit integer quantization described in Section 3.3. After dimensionality reduction, the first component of $\tilde{\mathbf{x}}$ corresponds to the principal axis. Thus, to further reduce the precision lost by quantization, we employ a mixed-precision decomposition by not quantizing the first component of $\tilde{\mathbf{x}}$ and the first row of both $\mathbf{A}$ and $\mathbf{B}$. Moreover, by pre-multiplying the projection matrix $\mathbf{W}_s$ with a random rotation matrix, we rotate the query point $\tilde{\mathbf{x}}$ at no extra cost before quantization. For re-quantizing $\mathbf{r} = \tilde{\mathbf{x}}^T\mathbf{A}$, since $\mathbf{A} = (\mathbf{X}\mathbf{W}_s)^\dagger\mathbf{Y}\mathbf{V}_r$ and $\mathbf{B} = \mathbf{V}_r^T$, we can again randomly rotate $\mathbf{V}_r$ beforehand at no extra cost.

We compute the 8-bit vector-matrix products $\mathbf{r}_{i32} = \tilde{\mathbf{x}}_{i8}^T\mathbf{A}_{i8}$ and $\hat{\mathbf{y}}_{i32} = \mathbf{r}_{i8}^T\mathbf{B}_{i8}$ efficiently on modern CPUs using VPDPBUSD instructions in the AVX-512 VNNI instruction set.[2] Since a VPDPBUSD instruction computes dot products of signed 8-bit integer vectors and unsigned 8-bit integer vectors, we store $\mathbf{A}_{i8}$ and $\mathbf{B}_{i8}$ as unsigned 8-bit integer matrices $\mathbf{A}_{i8}' = \mathbf{A}_{i8} + 128 \cdot \mathbf{1}_{s \times r} \in [\mathbb{Z}]_{256}^{d \times r}$ and $\mathbf{B}_{i8}' = \mathbf{B}_{i8} + 128 \cdot \mathbf{1}_{r \times m} \in [\mathbb{Z}]_{256}^{r \times m}$, and compute $\mathbf{r}_{i32} = \tilde{\mathbf{x}}_{i8}^T\mathbf{A}_{i8} = \tilde{\mathbf{x}}_{i8}^T\mathbf{A}_{i8}' - 128 \cdot \tilde{\mathbf{x}}_{i8}^T\mathbf{1}_{s \times r}$. The dimensionality reduction lowers the memory usage of `LoRANN` from $Ldr + rm$ to $Lsr + rm$ bytes.

## 4.3   GPU implementation

Hardware accelerators such as GPUs and TPUs can be used to speed up ANN search for queries that arrive in batches (Johnson et al., 2019; Zhao et al., 2020; Groh et al., 2022; Ootomo et al., 2023). The computational structure of RRR, consisting of vector-matrix multiplications, makes it easy to implement `LoRANN` for accelerators, and the low memory usage of RRR (see Section 7.1.2) makes it ideal for accelerators that typically have a limited amount of memory.

Given a query matrix $\mathbf{Q} \in \mathbb{R}^{|Q| \times d}$ with $|Q|$ queries, we need to compute the products $\mathbf{q}_i^T\mathbf{A}_{il}\mathbf{B}_{il}$ for all $i = 1, \ldots, |Q|$ and $l = 1, \ldots, w$. Here we denote by $\mathbf{A}_{il}$ and $\mathbf{B}_{il}$ the matrices $\mathbf{A}$ and $\mathbf{B}$ of the $l$th cluster in the set of $w$ clusters the $i$th query point is routed into. We compute the required products efficiently using one batched matrix multiplication $\mathbf{Q_T}\mathbf{A_T}\mathbf{B_T}$ by representing $\mathbf{Q}$ as a $|Q| \times 1 \times 1 \times d$ tensor $\mathbf{Q_T}$, the matrices $\mathbf{A}_{il}$ as one $|Q| \times w \times s \times r$ tensor $\mathbf{A_T}$, and the matrices

---

[2]`https://en.wikichip.org/wiki/x86/avx512_vnni`

$\mathbf{B}_{il}$ as one $|Q| \times w \times r \times M$ tensor $\mathbf{B_T}$, where $M$ is the maximum number of points in a single cluster. To avoid inefficiencies due to padding clusters with fewer than $M$ points, we use an efficient balanced $k$-means algorithm (de Maeyer et al., 2023) to ensure that the clusters are balanced such that the maximum difference in cluster sizes is $\Delta = 16$.

On GPUs, 8-bit integer multiplication is presently both less efficient and less supported than 16-bit floating-point multiplication. Therefore, we use 16-bit floating-point numbers to perform all computations on a GPU. However, the 8-bit quantization scheme can still be useful on accelerators by allowing bigger data sets to be indexed with limited memory.

The simple structure of our method allows it to be easily implemented using frameworks such as PyTorch, TensorFlow, and JAX. This enables LoRANN to support different hardware platforms with minimal differences between implementations. We write our GPU implementation in Python using JAX which uses XLA to compile and run the algorithm on a GPU or a TPU (Frostig et al., 2018).

## 5 Out-of-distribution queries

In the standard benchmark set-up (Li et al., 2019; Aumüller et al., 2020), a data set is randomly split into the corpus and a test set, i.e., the corpus and the queries are drawn from the same distribution. However, this assumption often does not hold in practice, for example in cross-modal search. Thus, there is recent interest (Simhadri et al., 2022; Jaiswal et al., 2022; Hyvönen et al., 2022) in the out-of-distribution (OOD) setting. For instance, in the Yandex-text-to-image data set, the corpus consists of images, and the queries are text; even though both the corpus and the queries are embedded in the same vector space, their distributions differ (see Figure 2 in Jaiswal et al., 2022).

Due to the regression formulation, the proposed method handles OOD queries by design. To verify this, we construct an index for a sample of 400K points of the Yandex OOD data set in four different scenarios: (1) the default version (LoRANN-query) uses $\{\mathbf{x}_i\}_{i=1}^{n}$, i.e., an $n = 400$K sample from the query distribution, as a global training set and selects the local training set $\{\mathbf{x}_i\}_{i \in J_l}$ as the global training set points that are routed into the $l$th cluster; (2) LoRANN-query-big is like LoRANN-query, but with $n = 1.2$M; (3) LoRANN-corpus uses the corpus $\{\mathbf{c}_j\}_{j=1}^{m}$ as a global training set, and selects the local training sets as $\{\mathbf{c}_j\}_{j \in J_l}$ like the default version; (4) LoRANN-corpus-local uses $\{\mathbf{c}_i\}_{j=1}^{m}$ as a global training set, but selects the local training sets as $\{\mathbf{c}_j\}_{j \in I_l}$, i.e., uses only the corpus points of the $l$th cluster to train the $l$th model. We can thus disentangle the effect of the choice of the global training set from the effect of using the points in the nearby clusters to train the RRR models.

The results are shown in Figure 1. The version trained on queries outperforms the version trained only on the corpus, especially in the case of no re-ranking. Furthermore, we can use larger training sets to increase the performance of LoRANN. Both LoRANN-query and LoRANN-corpus outperform LoRANN-corpus-local, indicating that selecting the local training sets as described in Section 3.2 improves the accuracy of the regression models. We assume that this is because of the larger and more representative training sets, even though they are not from the actual query distribution.

## 6 Related work

**Supervised ANN algorithms**   Learning-to-hash methods (Weiss et al., 2008; Norouzi and Fleet, 2011; Liu et al., 2012) optimize partitions in a supervised fashion using data-dependent hash functions. Other supervised methods include learning optimal partitions by approximating a balanced graph partitioning (Dong et al., 2020; Gupta et al., 2022; Gottesbüren et al., 2024) and interpreting partitions as multilabel classifiers (Hyvönen et al., 2022). These supervised methods are orthogonal to our approach since they define the learning problem as selecting a subset of corpus points via partitioning. In contrast, we propose a supervised score computation method for clustering-based or, more generally, for partition-based ANN algorithms.

**Product quantization**   The state-of-the-art clustering-based algorithms IVF-PQ (Jegou et al., 2011) and ScaNN (Guo et al., 2020) use quantization for data compression and score computation. They use a *quantizer* $q : \mathbb{R}^d \mapsto \mathcal{A}$ to map a point of the feature space to a value in a *codebook* $\mathcal{A}$. Given $\mathcal{A}$, they approximate the dissimilarity between the query point $\mathbf{x}$ and the corpus point $\mathbf{c}_j$ by $d(\mathbf{x}, q(\mathbf{c}_j))$, i.e., the dissimilarity between the query point and the codebook value corresponding

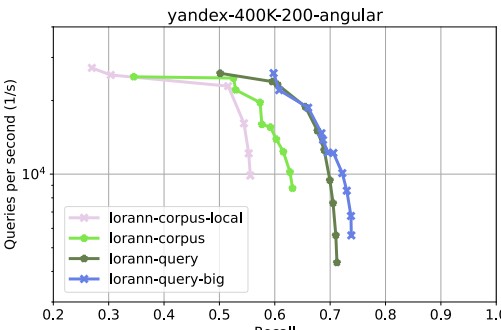 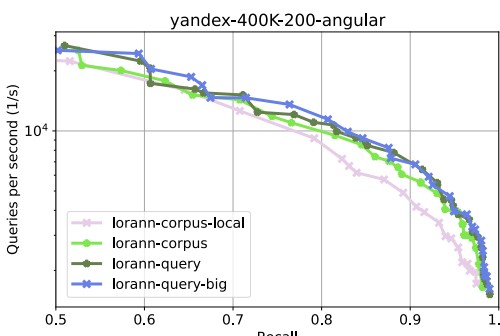

Figure 1: Recall vs. QPS on the Yandex T2I OOD data set (400K sampled corpus points) without (left) and with (right) the final re-ranking step. `LoRANN-query` is trained using a sample of 400K points from the query distribution as a training set, while `LoRANN-query-big` uses a sample of 1.2M points. `LoRANN-corpus` is trained using the corpus as a training set. `LoRANN-corpus-local` is trained using the corpus as a training set with only the cluster points as the local training sets of the reduced-rank regression models. It is beneficial to (1) use a sample from the actual query distribution as a training set and to (2) select the local training set by using also the points outside of the cluster as described in Section 3.2. The performance difference decreases when the final re-ranking step is introduced (requiring the original data set to be kept in memory).

to $\mathbf{c}_j$. Further, IVF-PQ and ScaNN quantize the residuals, i.e., the distances between corpus points and the cluster centroids, and use product quantization that decomposes the feature space into lower-dimensional subspaces and learns subquantizers in these subspaces. The code size, and thus the memory consumption, of PQ is directly proportional to the number of subquantizers.

# 7   Experiments

We use the ANN-benchmarks project (Aumüller et al., 2020) to run our experiments[3] and replicate its experimental set-up as closely as possible (see Appendix B for the description of the experimental set-up). We use $k = 100$ for all experiments and measure recall (the proportion of true $k$-nn found) versus queries per second (QPS). Additionally, due to the lack of modern high-dimensional embedding data sets in ANN-benchmarks, we include multiple new high-dimensional embedding data sets in our experiments; for a description of all the data sets, see Appendix C.

Note that, even though we demonstrated in Section 5 that `LoRANN` can adapt to the query distribution, there are no samples from the actual query distribution available for the benchmark data sets of this section. Thus, we follow the standard approach by using only the corpus $\{\mathbf{c}_j\}_{j=1}^m$ to train `LoRANN`.

## 7.1   Reduced-rank regression

We first compare the proposed score computation method, reduced-rank regression (RRR), against product quantization (PQ) for clustering-based ANN search.[4] We use RRR and PQ as scoring methods for an IVF index that partitions the corpus using $k$-means (see Section 2.2). We implement RRR using 8-bit integer quantization as described in Section 3.3 and compare against product quantization implemented in Faiss (Douze et al., 2024) with 4-bit integer quantization and fast scan (André et al., 2015). First, we compare the score computation methods at the optimal hyperparameters while keeping the clustering fixed, and then compare them at a fixed memory budget.

---

[3]`https://github.com/ejaasaari/lorann-experiments`

[4]Since ScaNN does not outperform Faiss-IVFPQ in our end-to-end-evaluation (Section 7.2.2) and both methods employ $k$-means for partitioning, the anisotropic quantization (Guo et al., 2020) used by ScaNN is unlikely to outperform the original PQ (Jegou et al., 2011).

### 7.1.1 Fixed clustering

To directly compare the score computation methods, in Figure 2 we present results where the IVF index (the partition defined by $k$-means clustering) is the same for both methods. For RRR, we use $r = 32$, while for PQ each vector is encoded with $d/2$ subquantizers for optimal performance. RRR outperforms PQ on seven out of the eight data sets; for complete results, see Appendix D.1.

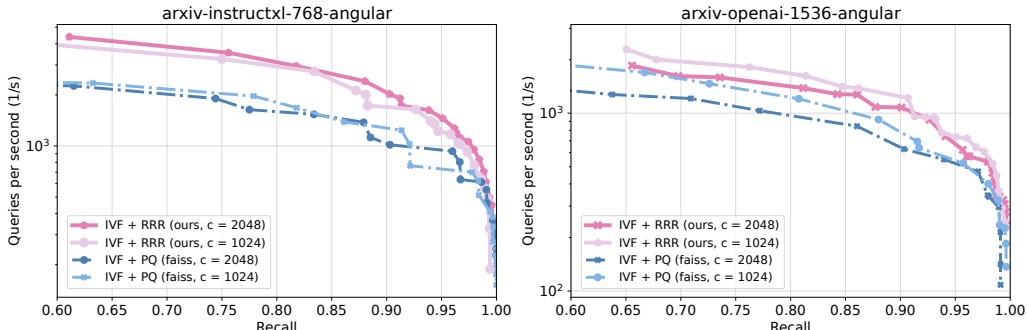

Figure 2: Performance comparison between RRR and PQ. The $k$-means clustering (IVF index) is kept constant to directly compare the effect of the score computation method (here $c$ denotes the number of clusters). The proposed score computation method outperforms the baseline method (PQ).

### 7.1.2 Memory usage

In Figure 3, we compare the performance of RRR (IVF+RRR) and PQ (IVF+PQ) by varying the rank parameter $r$ for RRR and the code size for PQ such that $b$, bytes per vector, is similar for both. For all values of $b$, RRR outperforms PQ which is a typical choice in memory-limited use cases. Note that RRR with $b \approx 16$ outperforms PQ even with $b \approx 64$ on all data sets; for the full results, see Appendix D.2. The results are similar when no final re-ranking step is used (Appendix D.3).

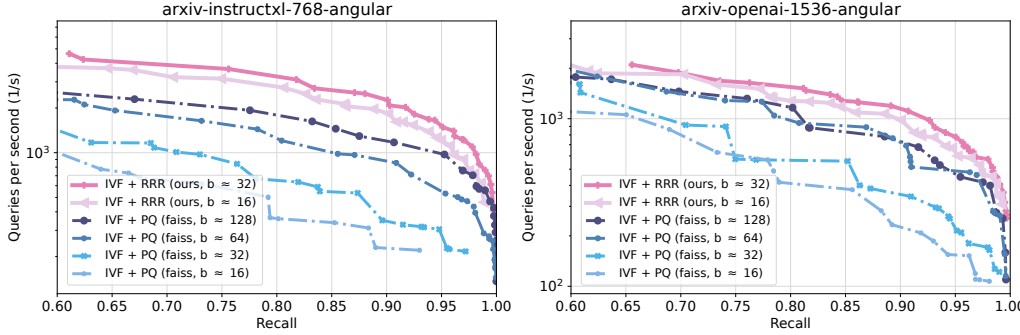

Figure 3: Performance comparison of RRR and PQ at different levels of memory usage. We vary the rank parameter $r$ for RRR and the code size for PQ such that $b$, bytes per vector, is similar for both. RRR@($b \approx 16$) outperforms even PQ@($b \approx 128$) which uses eight times as much memory.

### 7.2 LoRANN

In this section, we measure the end-to-end performance of LoRANN (see Section 4). We first perform an ablation study on the components of LoRANN (Section 7.2.1), and then perform an end-to-end evaluation of LoRANN against the state-of-the-art ANN libraries in both the CPU setting (Section 7.2.2) and the GPU setting (Section 7.2.3).

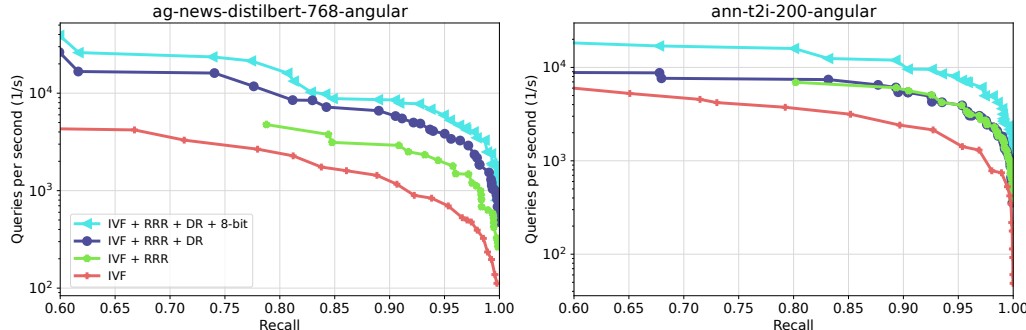

Figure 4: `LoRANN` ablation study. On the high-dimensional ($d = 768$) data set (left), all the components improve the performance of `LoRANN`. On the lower-dimensional ($d = 200$) data set (right), all the components except dimensionality reduction (DR) improve performance.

### 7.2.1 Ablation study

In Figure 4, we study the effect of the different components of `LoRANN` on its performance (for the full results, see Appendix E.1). As the baseline, we use an IVF index. Adding the score computation step via RRR significantly improves performance on all the data sets.

Dimensionality reduction (DR) is beneficial on higher-dimensional data sets with moderate-sized corpora: if the number of points in a cluster is lower than the dimension, then the first vector-matrix product in the computation $(\mathbf{x}^T \mathbf{A})\mathbf{B}$ of the local reduced-rank regression models will be more expensive. For large data sets with high-dimensional data, this effect decreases. Dimensionality reduction also does not improve performance on the lower-dimensional data sets ($d \leq 300$). Incorporating 8-bit quantization improves not only the memory usage but also the query latency.

In Appendix E.2, we study the effect of varying the rank $r$ of the parameter matrices. We find that increasing $r$ from 32 to 64 has little effect, while $r = 16$ performs worse for high-dimensional data (but can be used to further decrease memory usage). For all of our other experiments, we use $r = 32$.

### 7.2.2 CPU evaluation

As the baseline methods, we choose four leading graph implementations, HNSW, GLASS[5], QSG-NGT, and PyNNDescent (Dong et al., 2011), two leading product quantization implementations, Faiss-IVFPQ (fast scan) and ScaNN, and the leading tree implementation MRPT (Hyvönen et al., 2016). See Figure 5 for the results and Appendix E.3 for the results on all 16 data sets. Three trends emerge: (1) `LoRANN` outperforms the product quantization methods on all data sets except glove-200-angular. (2) `LoRANN` performs better in the high-dimensional regime: it outperforms all the other methods except GLASS and QSG-NGT on all but the lower-dimensional ($d \leq 200$) data sets. (3) Compared to graph methods, `LoRANN` performs better at the lower recall levels: QPS-recall curves of `LoRANN` and GLASS cross between 80% and 99% on most of the data sets. On 8 of the 16 data sets, `LoRANN` has better or similar performance as QSG-NGT at all recall levels.

Furthermore, in Appendix E.4, we demonstrate that in general `LoRANN` has faster index construction times than the graph-based methods.

### 7.2.3 GPU evaluation

We compare `LoRANN` against GPU implementations of IVF and IVF-PQ in both Faiss (Douze et al., 2024) and the NVIDIA RAFT library[6]. In addition, we compare against a state-of-the-art GPU graph algorithm CAGRA (Ootomo et al., 2023) implemented in RAFT. All algorithms receive all test queries as one batch of size 1000. See Figure 6 for representative results, and Appendix E.5 for the complete results. `LoRANN` outperforms the other methods on seven out of the nine high-dimensional ($d > 300$) data sets. For $d \leq 300$, CAGRA has the best performance.

---

[5]An efficient HNSW implementation with quantization: `https://github.com/zilliztech/pyglass`
[6]`https://github.com/rapidsai/raft`

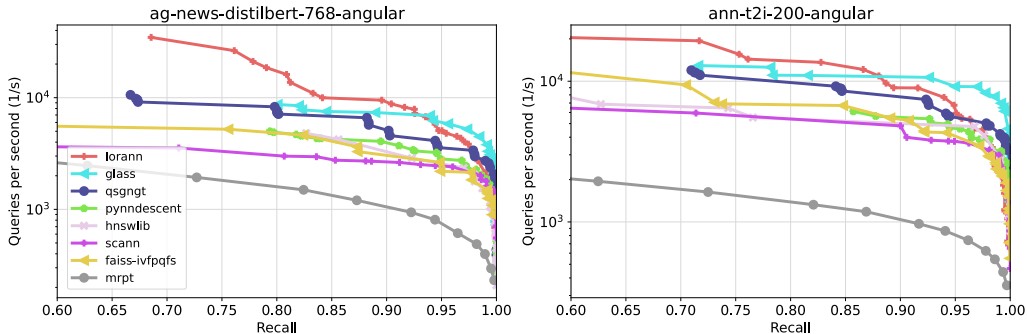

Figure 5: CPU comparison. The QPS-recall curves of LoRANN and the leading graph library GLASS cross at the 95% (left) and at the 90% recall level (right), indicating that LoRANN is the fastest method at the lower recall levels, and GLASS at the higher recall levels.

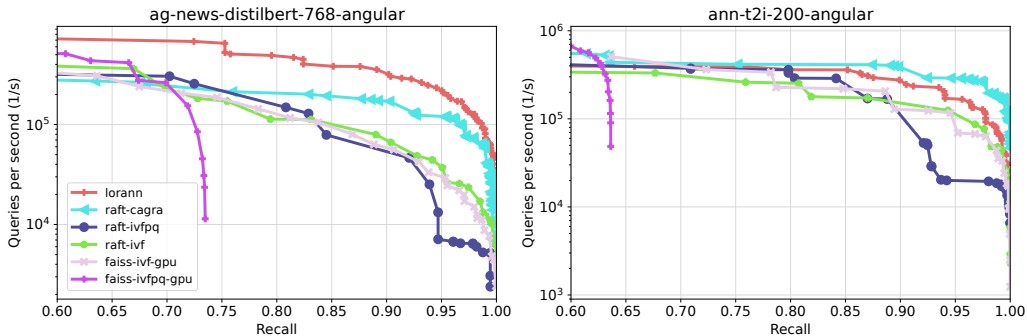

Figure 6: GPU comparison. On the high-dimensional ($d = 768$) data set (left), LoRANN is the fastest, and on the lower-dimensional ($d = 200$) data set (right), the graph method CAGRA is the fastest.

To demonstrate the ease of implementation of our algorithm for new hardware platforms, in Appendix E.5 we implement our method for Apple silicon. We show using the M2 Pro SoC that LoRANN can take advantage of the M2 GPU and its unified memory architecture to achieve faster queries.

## 8  Discussion

In this article, we show that an elementary statistical method, reduced-rank regression (RRR), is surprisingly efficient for score computation in clustering-based ANN search. Since RRR outperforms product quantization (PQ) at fixed memory consumption while being simpler to implement efficiently, we recommend using it in regimes where PQ is traditionally used, e.g., when the memory usage is a limiting factor (Douze et al., 2024). While the experiments of the article are performed in the standard in-memory setting, the simple structure and the small memory footprint of the proposed method suggest scalability for larger data sets that do not fit into the main memory. In particular, hybrid solutions that store the corpus on an SSD (Jayaram Subramanya et al., 2019; Ren et al., 2020; Chen et al., 2021) and the distributed setting where the corpus and the index are distributed over multiple machines (Deng et al., 2019; Gottesbüren et al., 2024) are promising research directions.

**Limitations**   Since reaching the highest recall levels ($> 90\%$ for $k = 100$) requires exploring many clusters, graph methods are usually more efficient than clustering-based methods in this regime. Furthermore, low-dimensional data sets (e.g., $d < 100$) also require that the rank $r$ of the parameter matrix is reasonably high. Thus, reduced-rank regression is not efficient for low-dimensional data sets for which the proportion $r/d$ is too large. The proposed score computation method is also only applicable to inner product-based dissimilarity measures. However, the multivariate regression formulation of Section 3.1 can be extended for other dissimilarity measures.

## Acknowledgments

This work has been supported by the Research Council of Finland (grant #345635 and the Flagship programme: Finnish Center for Artificial Intelligence FCAI) and the Jane and Aatos Erkko Foundation (BioDesign project, grant #7001702). We acknowledge the computational resources provided by the Aalto Science-IT Project from Computer Science IT. We thank the anonymous reviewers for their valuable feedback.

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

# A Geometric intuition

Observe that the reduced-rank regression solution is different from the most obvious low-rank approximation of the OLS solution computed via an SVD of the matrix $\mathbf{C}$. In Section 5, we verify experimentally using a real-world data set that a more accurate model can indeed be trained by factorizing $\mathbf{XC}^T$ instead of $\mathbf{C}$. These two solutions are equivalent only in the special case where $\mathbf{X} = \mathbf{C}$. In this special case, computing the predictions of the reduced-rank regression model, denoted by $\hat{\mathbf{y}}_{\text{RRR}} := \mathbf{x}^T \hat{\boldsymbol{\beta}}_{\text{RRR}}$, is equivalent to projecting both the query point $\mathbf{x}$ and the cluster points $\{\mathbf{c}_j\}_{j \in I_j}$ onto the subspace spanned by the first $r$ eigenvectors of $\mathbf{K} := \mathbf{C}^T \mathbf{C}$, and computing the inner products in this $r$-dimensional subspace. If the cluster points are centered, the eigenvectors of $\mathbf{K}$ are the principal axes of the cluster points, and $\frac{1}{m_l - 1} \mathbf{K}$ can be interpreted as the sample covariance matrix. However, unlike in principal component analysis (PCA), data should not be centered when estimating the inner products, since the inner products are not invariant with respect to translation.

Let $\mathbf{W}_r \in \mathbb{R}^{d \times r}$ be a matrix containing the first $r$ eigenvectors of $\mathbf{K}$ as columns, and denote by $\tilde{\mathbf{x}} := \mathbf{W}_r^T \mathbf{x}$ the $r$-dimensional projection of the query point, by $\tilde{\mathbf{C}} := \mathbf{CW}_r$ an $m_l \times r$-matrix containing the projected cluster points, and by $\tilde{\mathbf{y}} := \tilde{\mathbf{x}}^T \tilde{\mathbf{C}}^T$ an $1 \times m_l$-matrix containing the inner products between the projected query and the projected cluster points. Using this notation, we have the following result:

**Theorem 1.** *If* $\mathbf{X} = \mathbf{C}$*, then* $\tilde{\mathbf{y}} = \hat{\mathbf{y}}_{RRR}$*.*

*Proof.* Denote $\tilde{\boldsymbol{\beta}} := \mathbf{W}_r \mathbf{W}_r^T \mathbf{C}$. Now $\tilde{\mathbf{y}}$ can be written as $\tilde{\mathbf{y}} = \mathbf{x}^T \tilde{\boldsymbol{\beta}}$. To prove the result, it suffices to show that $\tilde{\boldsymbol{\beta}} = \hat{\boldsymbol{\beta}}_{\text{RRR}}$. Denote the singular value decomposition of $\mathbf{C}^T$ by $\mathbf{C}^T = \mathbf{U}\boldsymbol{\Sigma}\mathbf{V}^T$. Since $\mathbf{X} = \mathbf{C}$, $\mathbf{Y} = \mathbf{CC}^T$, and, consequently, the right singular vectors of $\mathbf{C}^T$ are the eigenvectors of $\mathbf{Y}$. Thus, the reduced-rank regression solution is

$$\hat{\boldsymbol{\beta}}_{\text{RRR}} = \mathbf{C}^T \mathbf{V}_r \mathbf{V}_r^T,$$

where $\mathbf{V}_r$ is $m \times r$-matrix containing the first $r$ columns of $\mathbf{V}$.

The columns of the matrix $\mathbf{U}$—i.e., the left singular vectors of $\mathbf{C}^T$—are the eigenvectors of $\mathbf{K} = \mathbf{C}^T \mathbf{C}$. Thus, $\mathbf{W}_r = \mathbf{U}_r$, and $\tilde{\boldsymbol{\beta}}$ can be written as

$$\tilde{\boldsymbol{\beta}} = \mathbf{U}_r \mathbf{U}_r^T \mathbf{C},$$

where we denote by $\mathbf{U}_r$ a $d \times r$-matrix containing the first $r$ columns of the matrix $\mathbf{U}$. Since $\mathbf{U}^T \mathbf{C}^T = \boldsymbol{\Sigma}\mathbf{V}^T$, also $\mathbf{U}_r^T \mathbf{C}^T = \boldsymbol{\Sigma}_r \mathbf{V}_r^T$, and since $\mathbf{C}^T \mathbf{V} = \mathbf{U}\boldsymbol{\Sigma}$, also $\mathbf{C}^T \mathbf{V}_r = \mathbf{U}_r \boldsymbol{\Sigma}_r$. Using these identities, we have

$$\tilde{\boldsymbol{\beta}} = \mathbf{U}_r \mathbf{U}_r^T \mathbf{C}^T = \mathbf{U}_r \boldsymbol{\Sigma}_r \mathbf{V}_r^T = \mathbf{C}^T \mathbf{V}_r \mathbf{V}_r^T = \hat{\boldsymbol{\beta}}_{\text{RRR}},$$

which completes the proof. $\qquad\square$

However, in the general case where $\mathbf{X} \neq \mathbf{C}$, i.e., the training set of the $l$th model is selected as the training queries that are routed into the $l$th cluster (for which $l \in \tau(\mathbf{x}_i)$), the matrix $\mathbf{X}$ also affects the right singular vectors of the output matrix $\mathbf{Y} = \mathbf{XC}^T$. Hence, there is no simple geometric interpretation for the reduced-rank regression solution $\hat{\boldsymbol{\beta}}_{\text{RRR}}$ in the $d$-dimensional feature space.

# B    Experimental set-up

We use the ANN-benchmarks project[7] (Aumüller et al., 2020) to run our experiments and replicate its set-up as close as possible: our experiments are performed on AWS `r6i.4xlarge` instances with Intel Xeon 8375C (Ice Lake) processors and hyperthreading disabled, and all algorithms are run using a single core. For our GPU experiments, we use an AWS `g5.2xlarge` instance with an NVIDIA A10G GPU (24 GB VRAM) and a `mac2-m2pro.metal` instance with an Apple M2 Pro SoC.

We use $k = 100$ for all experiments and measure the recall (the proportion of true $k$-nn found) versus queries per second (QPS). For all data sets, we use a separate test set of 1000 queries, and each hyperparameter combination is benchmarked 5 times, from which the lowest achieved query latency is recorded. Additionally, due to the lack of modern high-dimensional embedding data sets in ANN-benchmarks, we include multiple new high-dimensional data sets in our experiments; for a list of all the data sets, see Appendix C. To account for the new data sets, we increase the sizes of the considered hyperparameter grids in applicable cases.

For `LoRANN` experiments (Section 7.2), we use the following hyperparameter grid:

- Number of clusters $L$: 1024, 2048, 4096
- Reduced dimension $s$: 64, 128, 192
- Rank $r$: 32 (for GPU, $r$: 24)
- Clusters to search $w$: 8, 16, 24, 32, 48, 64, 128, 256
- Points to re-rank $t$: 100, 200, 400, 800, 1200, 1600, 2400, 3200

For experiments on just reduced-rank regression (Section 7.1), we use the same hyperparameter grid for $L, w$, and $t$ as above and do not use dimensionality reduction.

For all hyperparameters, see `https://github.com/ejaasaari/lorann-experiments`.

---

[7] `https://github.com/erikbern/ann-benchmarks`

## C  Data sets

In addition to data sets from ANN-benchmarks (fashion-mnist, gist, glove, mnist, sift), we include high-dimensional neural network embedding data sets that are more representative of those encountered in modern machine learning applications and encompass dimensionalities ranging from 200 to 1536. The full list of data sets along with their sizes and the distance metrics used is given in Table 1.

Table 1: Data sets used in the experiments.

| Data set (model) | type | corpus size | dim | distance | license |
|---|---|---|---|---|---|
| ag-news (DistilBERT)[1] | text embedding | 120 000 | 768 | angular | CC BY 4.0 |
| ag-news (MiniLM)[1] | text embedding | 120 000 | 384 | angular | CC BY 4.0 |
| ann-arxiv (E5-base)[2] | text embedding | 2 288 300 | 768 | angular | Apache 2.0 |
| ann-t2i (ResNext)[3] | image embedding | 1 000 000 | 200 | angular | Apache 2.0 |
| arxiv (OpenAI Ada)[4] | text embedding | 319 224 | 1536 | angular | CC0 1.0 |
| arxiv (InstructXL)[5] | text embedding | 2 254 000 | 768 | angular | N/A |
| fashion-mnist[6] | raw image | 60 000 | 784 | euclidean | MIT |
| fasttext-wiki[7] | word embedding | 1 000 000 | 300 | euclidean | CC BY-SA 3.0 |
| gist[8] | image descriptor | 1 000 000 | 960 | euclidean | CC0 1.0 |
| glove[9] | word embedding | 1 193 514 | 200 | angular | Apache 2.0 |
| mnist[10] | raw image | 60 000 | 784 | euclidean | CC BY-SA 3.0 |
| sift[8] | image descriptor | 1 000 000 | 128 | euclidean | CC0 1.0 |
| wiki (GTE-small)[11] | text embedding | 224 482 | 384 | angular | MIT |
| wiki (OpenAI Ada)[11] | text embedding | 224 482 | 1536 | angular | MIT |
| wolt (clip-ViT)[12] | image embedding | 1 720 611 | 512 | angular | N/A |
| yandex T2I 5M[13] | text/image | 5 000 000 | 200 | angular | CC BY 4.0 |

[1] `https://data.dtu.dk/articles/dataset/Pretrained_sentence_BERT_models_AG_News_embeddings/21286923`
[2] `https://huggingface.co/datasets/unum-cloud/ann-arxiv-2m`
[3] `https://huggingface.co/datasets/unum-cloud/ann-t2i-1m`
[4] `https://www.kaggle.com/datasets/awester/arxiv-embeddings`
[5] `https://huggingface.co/datasets/Qdrant/arxiv-abstracts-instructorxl-embeddings`
[6] `https://github.com/zalandoresearch/fashion-mnist`
[7] `https://fasttext.cc/docs/en/english-vectors.html`
[8] `http://corpus-texmex.irisa.fr/`
[9] `https://nlp.stanford.edu/projects/glove/`
[10] `https://yann.lecun.com/exdb/mnist/`
[11] `https://huggingface.co/datasets/Supabase/wikipedia-en-embeddings`
[12] `https://huggingface.co/datasets/Qdrant/wolt-food-clip-ViT-B-32-embeddings`
[13] `https://big-ann-benchmarks.com/neurips23.html`

# D Reduced-rank regression experiments

## D.1 Fixed clustering

In this section, we present the complete evaluation of reduced-rank regression in comparison to product quantization when the $k$-means clustering is kept fixed. For discussion, refer to Section 7.1.1.

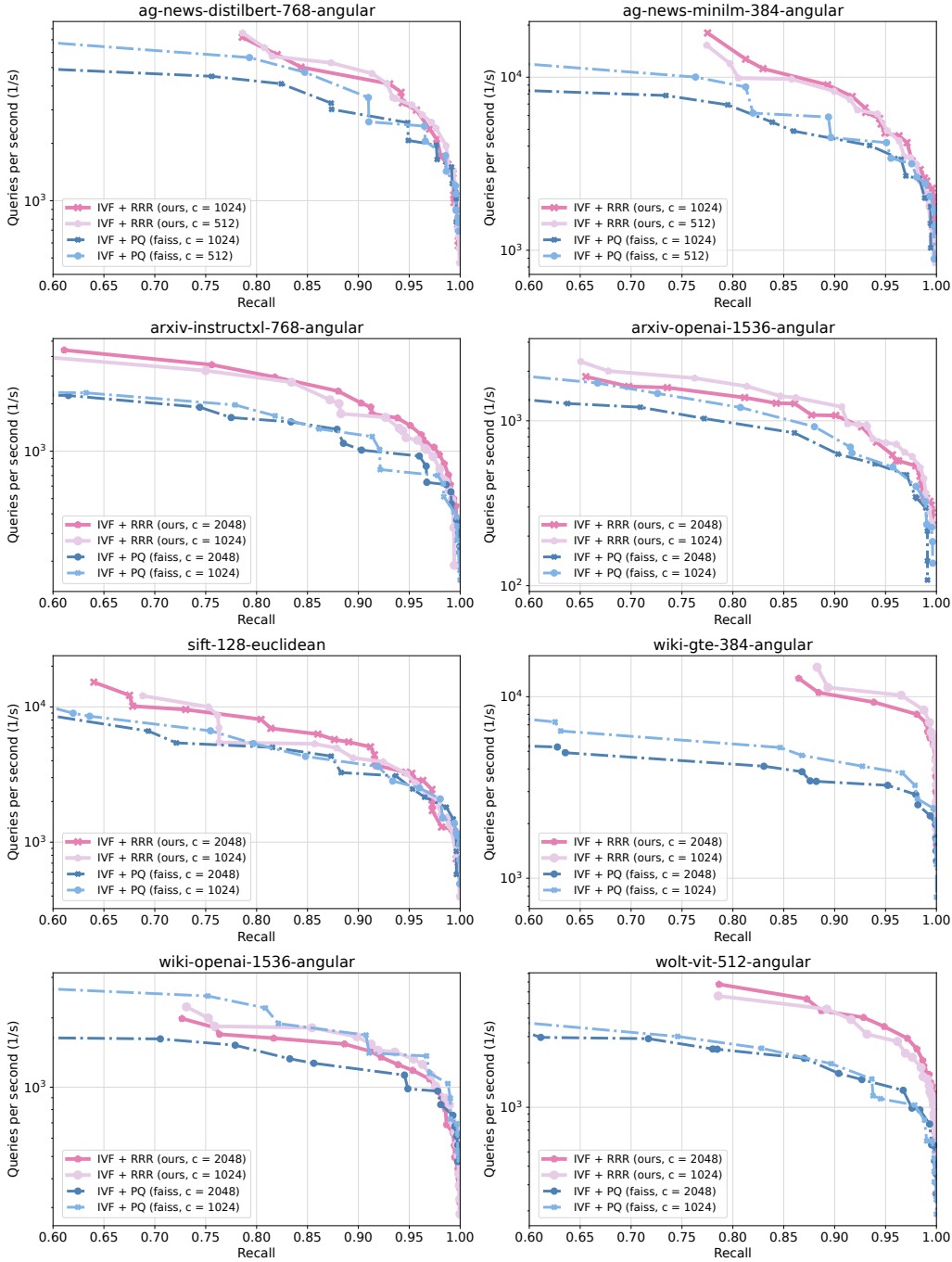

## D.2 Memory usage

In this section, we present the complete evaluation of reduced-rank regression in comparison to product quantization for different memory consumptions. For discussion, refer to Section 7.1.2.

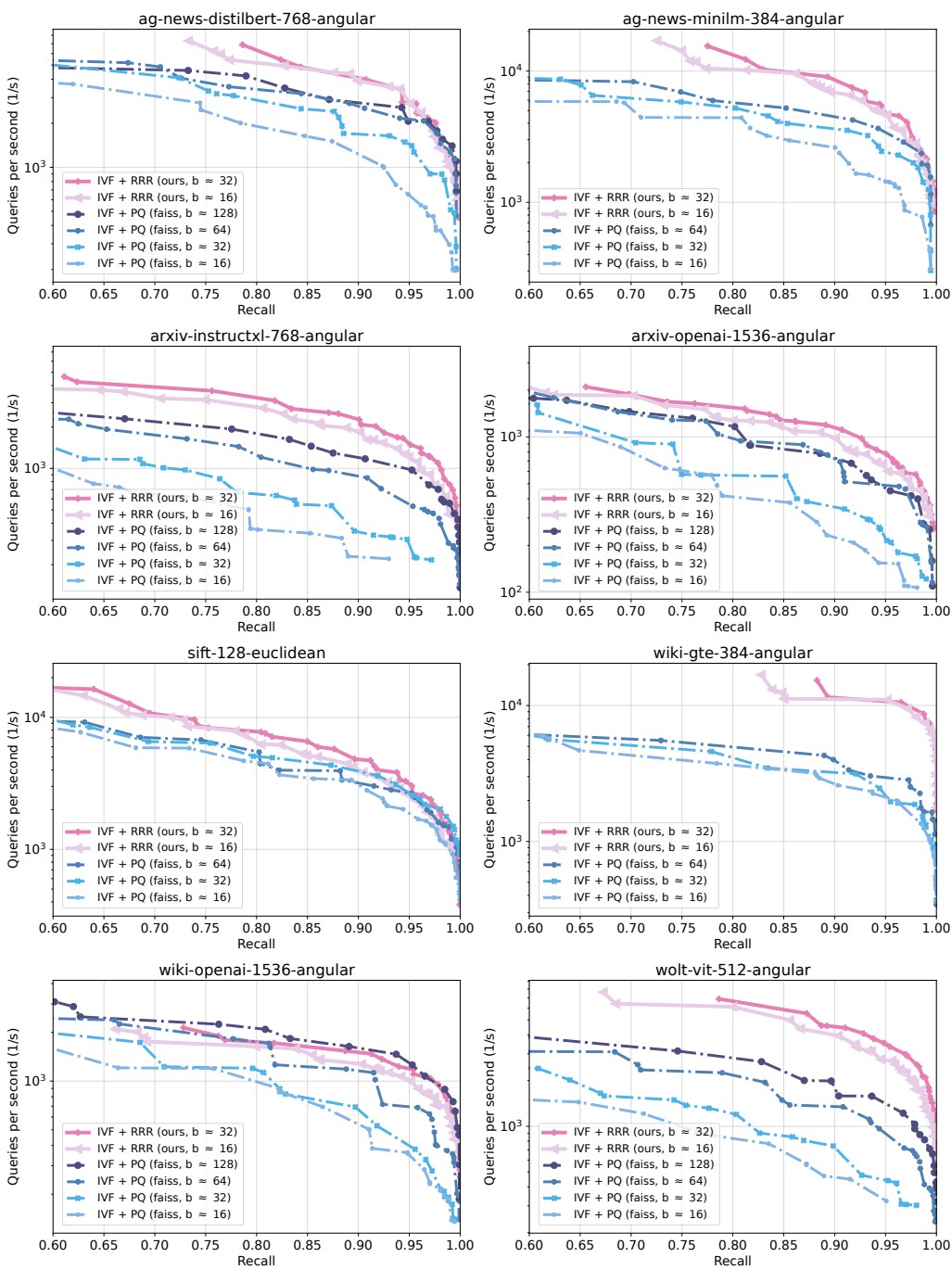

## D.3 Memory usage (no re-ranking)

In this section, we present the evaluation of reduced-rank regression in comparison to product quantization for different memory consumptions when no final re-ranking step is used. In this scenario, the original data set does not have to be kept in memory.

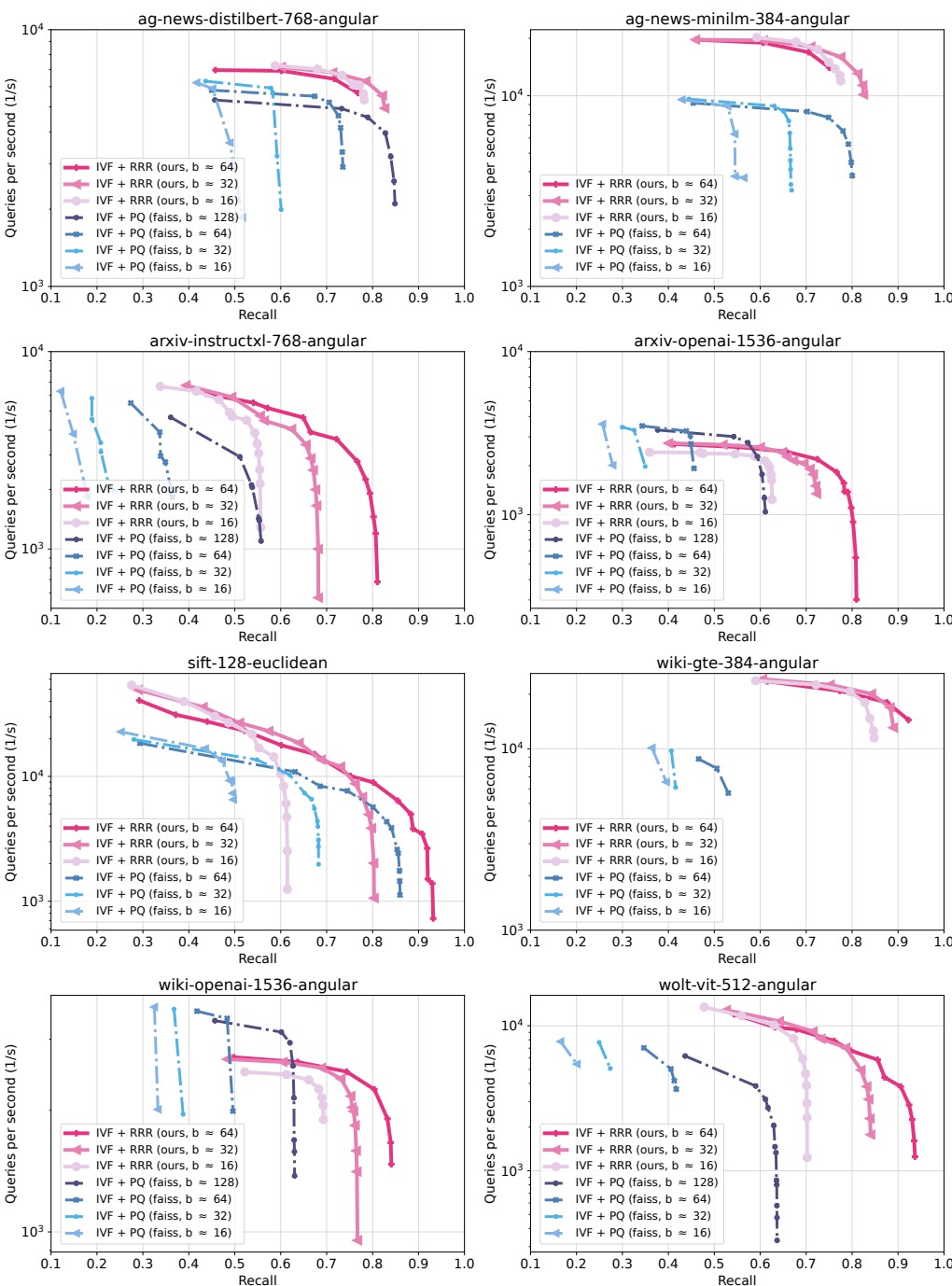

# E   LoRANN experiments

## E.1   Ablation study

In this section, we study the effect of the components (reduced-rank regression, dimensionality reduction, 8-bit quantization) of `LoRANN` on its performance. For discussion, refer to Section 7.2.1.

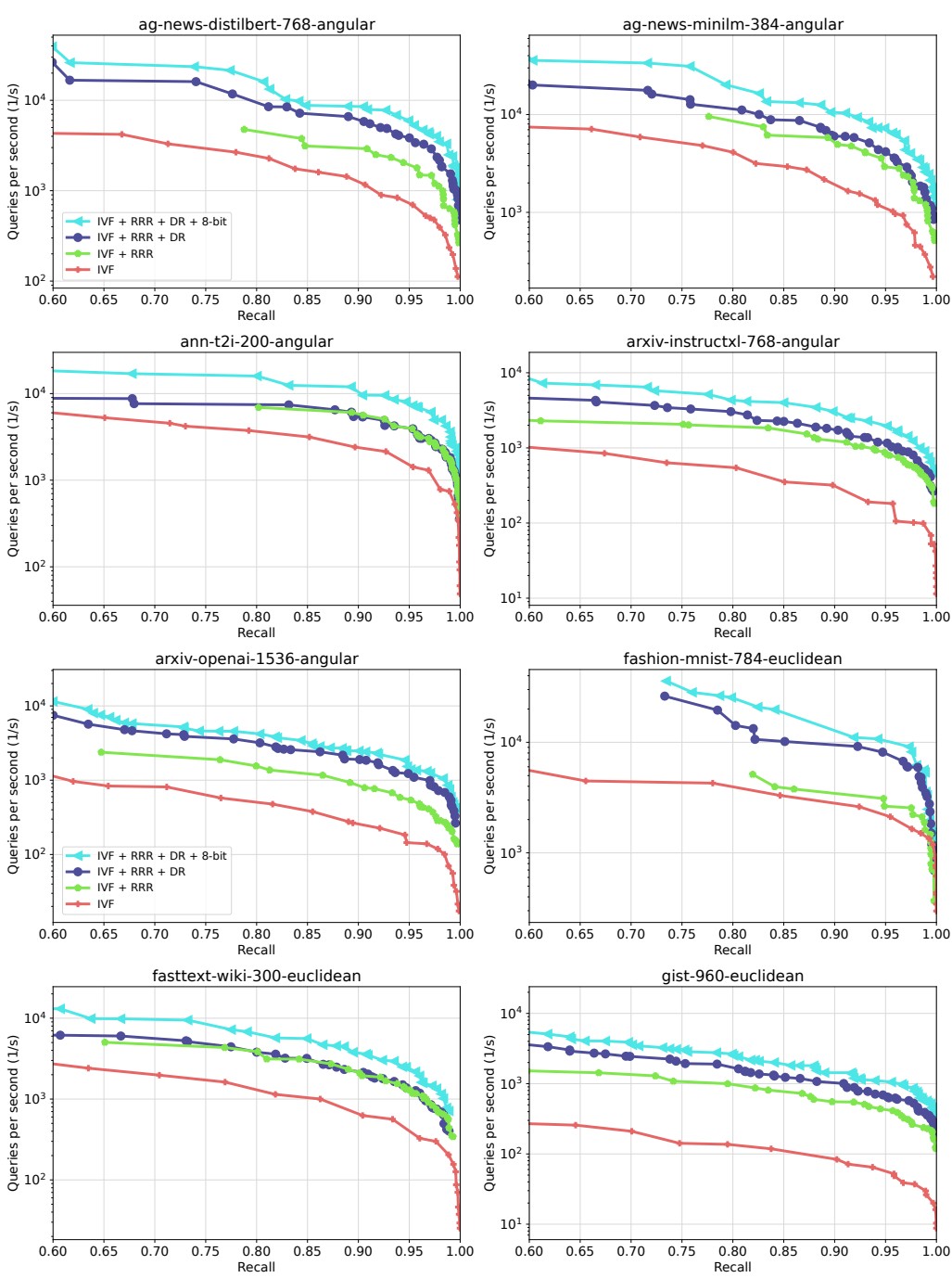

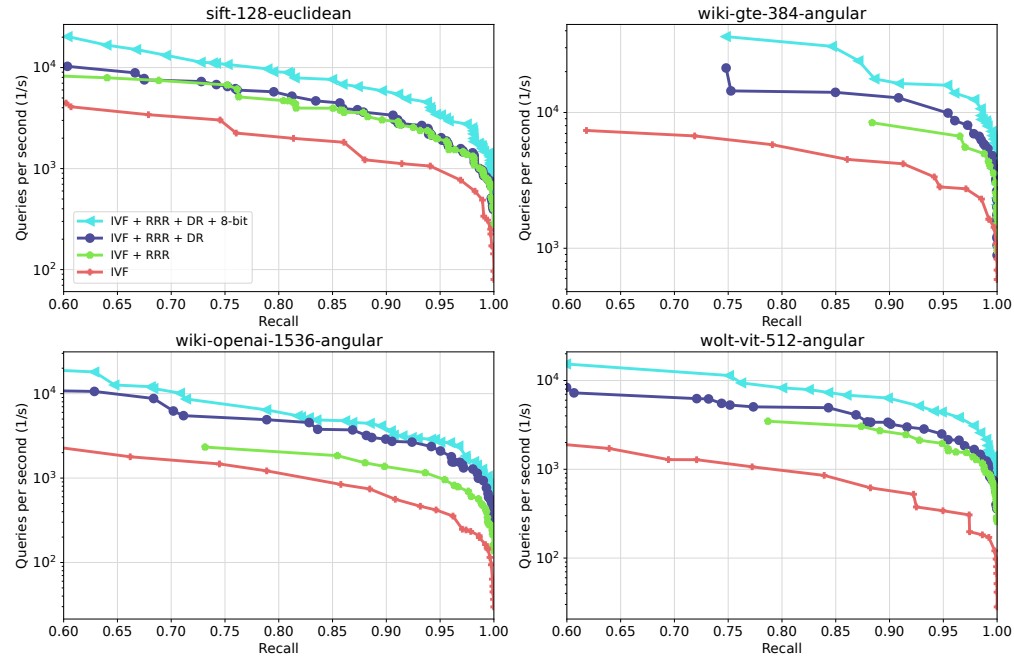

## E.2 Effect of the rank

In Figure 7, we study the impact of the rank $r$ of the parameter matrices $\hat{\boldsymbol{\beta}}_{\text{RRR}}$. We find that LoRANN is robust with respect to the choice of $r$: increasing $r$ from 32 to 64 has little effect (when the final re-ranking step is used), while $r = 16$ performs only slightly worse for high-dimensional data. In our experiments, we use $r = 32$, but $r = 16$ can be used to decrease the memory consumption, and $r = 64$ can be used to achieve higher recall if the final re-ranking step is not used.

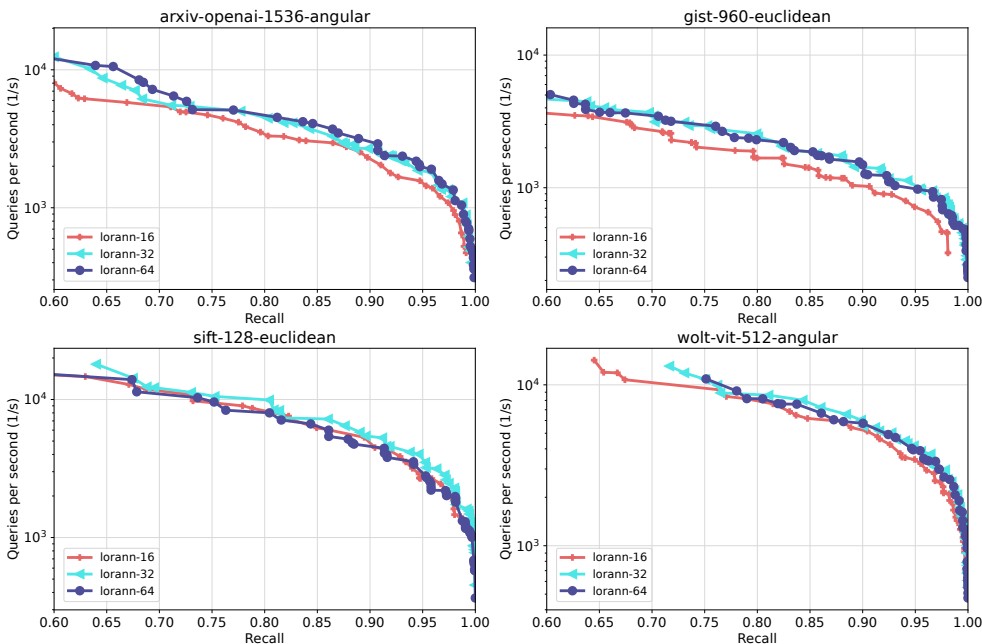

Figure 7: Effect of the rank $r$ of the parameter matrices $\hat{\boldsymbol{\beta}}_{\text{RRR}}$ in LoRANN.

## E.3 CPU Evaluation

In this section, we present the complete evaluation of LoRANN in the CPU setting in comparison to other ANN algorithms. For discussion, refer to Section 7.2.2.

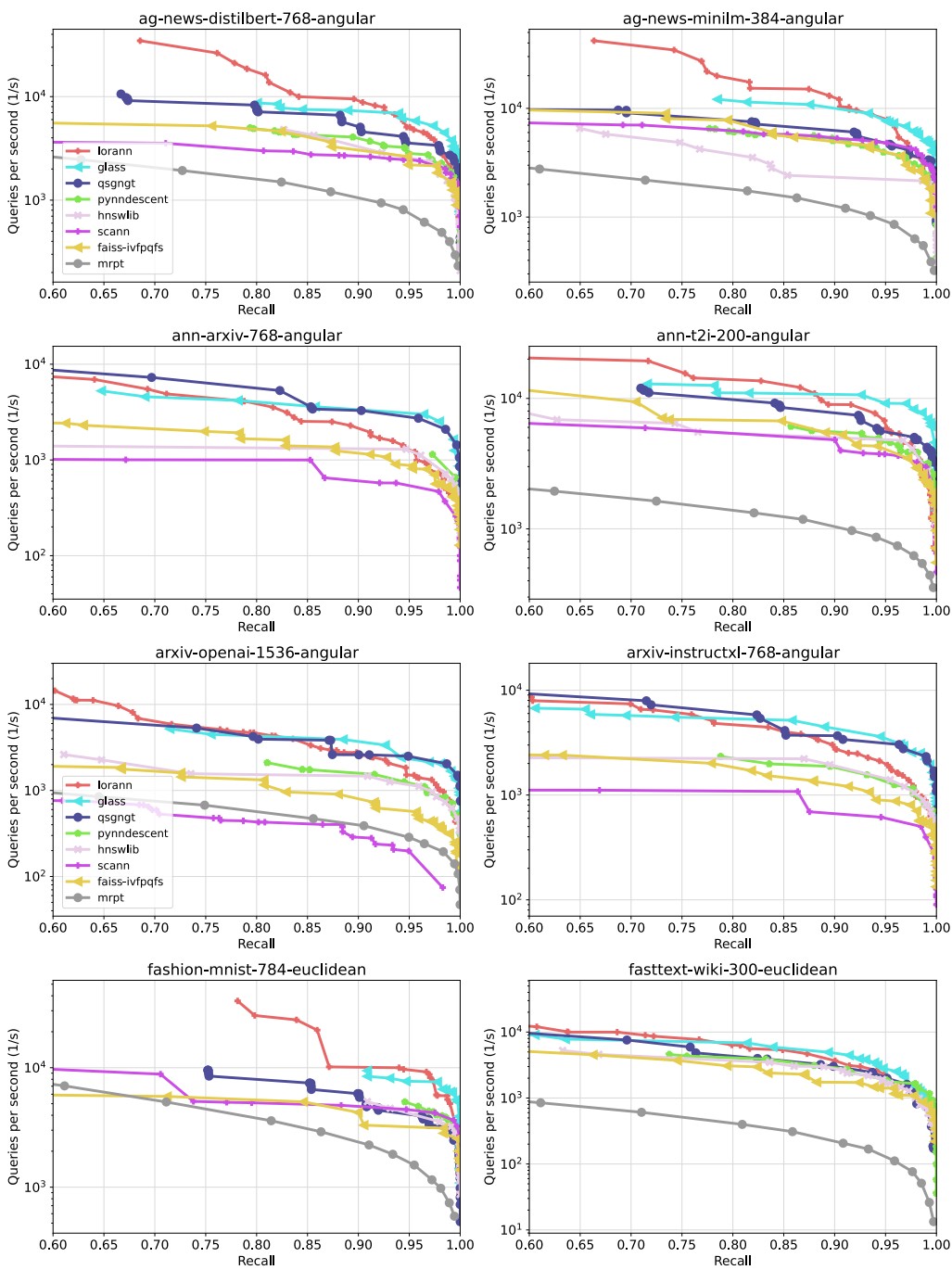

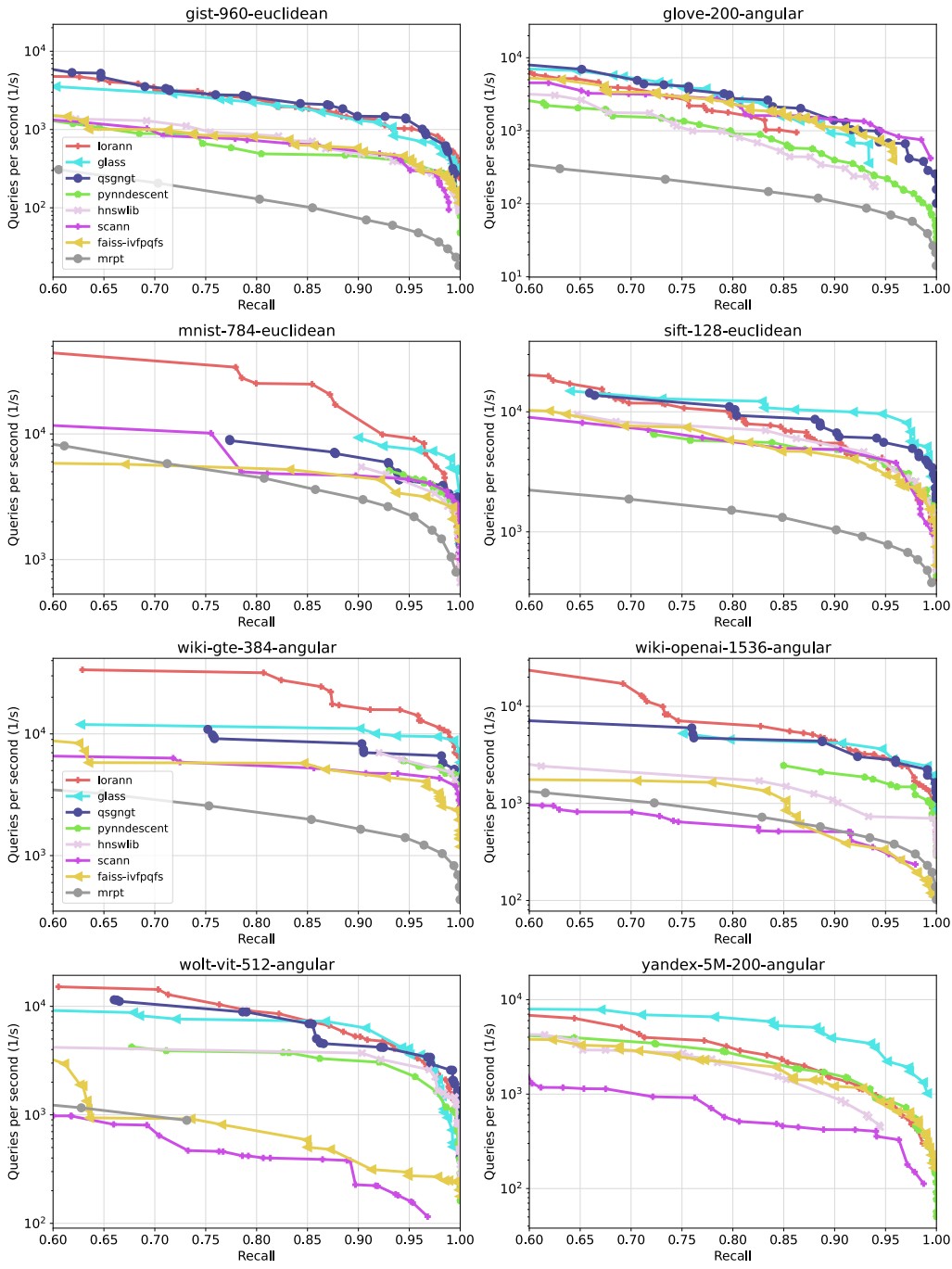

### E.4 Index construction time

For all libraries, we measure the index construction time on a single CPU core (for LoRANN, the index construction parallelizes easily as all local cluster models can be computed independently). Table 2 shows index construction times for all data sets and compared methods at $90\%$ recall for $k = 100$ at the optimal hyperparameters (a dash means that the algorithm either did not reach the necessary recall or ran out of memory on the data set).

LoRANN has slower index construction times than IVFPQfs, and similar index construction times as ScaNN. In general, LoRANN has faster index construction times than the graph methods: QSG-NGT has extremely slow index construction times, HNSW has slower index construction times than LoRANN on all data sets except ann-t2i, and GLASS is slower on all data sets except two lower-dimensional data sets (ann-t2i and fasttext-wiki).

We note that our implementation of LoRANN has not been optimized with respect to index construction time, and could be improved with further sampling and more approximate SVD computations.

Table 2: Index construction times (seconds) at $90\%$ recall for optimal hyperparameters.

| data set | IVFPQfs | LoRANN | ScaNN | NNDesc. | QSGNGT | Glass | HNSW |
|---|---|---|---|---|---|---|---|
| ag-distilbert | 29 | 26 | 66 | 56 | 1687 | 73 | 104 |
| ag-minilm | 15 | 18 | 34 | 47 | 1763 | 46 | 203 |
| ann-arxiv | 269 | 1442 | 1160 | 3988 | 16521 | 4241 | 7421 |
| ann-t2i | 34 | 916 | 125 | 238 | 3366 | 304 | 842 |
| arxiv-instructxl | 79 | 1189 | 1259 | 1041 | 18781 | 2956 | 6779 |
| arxiv-openai | 77 | 112 | 425 | 208 | 16369 | 405 | 926 |
| fashion-mnist | 15 | 6 | 30 | 32 | 1254 | 22 | 42 |
| fasttext-wiki | 60 | 1012 | – | 1241 | 3025 | 752 | 1414 |
| gist | 979 | 305 | 686 | 1203 | 8610 | 2258 | 6631 |
| glove | 49 | – | 155 | 1958 | 14259 | 1728 | 4412 |
| mnist | 15 | 6 | 31 | 32 | 1249 | 22 | 43 |
| sift | 24 | 91 | 32 | 361 | 2785 | 405 | 851 |
| wiki-gte | 24 | 34 | 39 | 59 | 2060 | 61 | 138 |
| wiki-openai | 362 | 67 | 296 | 137 | 5085 | 239 | 506 |
| wolt-vit | 286 | 464 | 563 | 1103 | 8327 | 874 | 1982 |
| yandex-5M | 589 | 2444 | 612 | 5218 | – | 2884 | 6599 |

## E.5 GPU evaluation

**NVIDIA GPU**   In this section, we present the full evaluation of LoRANN in the GPU setting in comparison to other GPU ANN methods. For discussion, refer to Section 7.2.3.

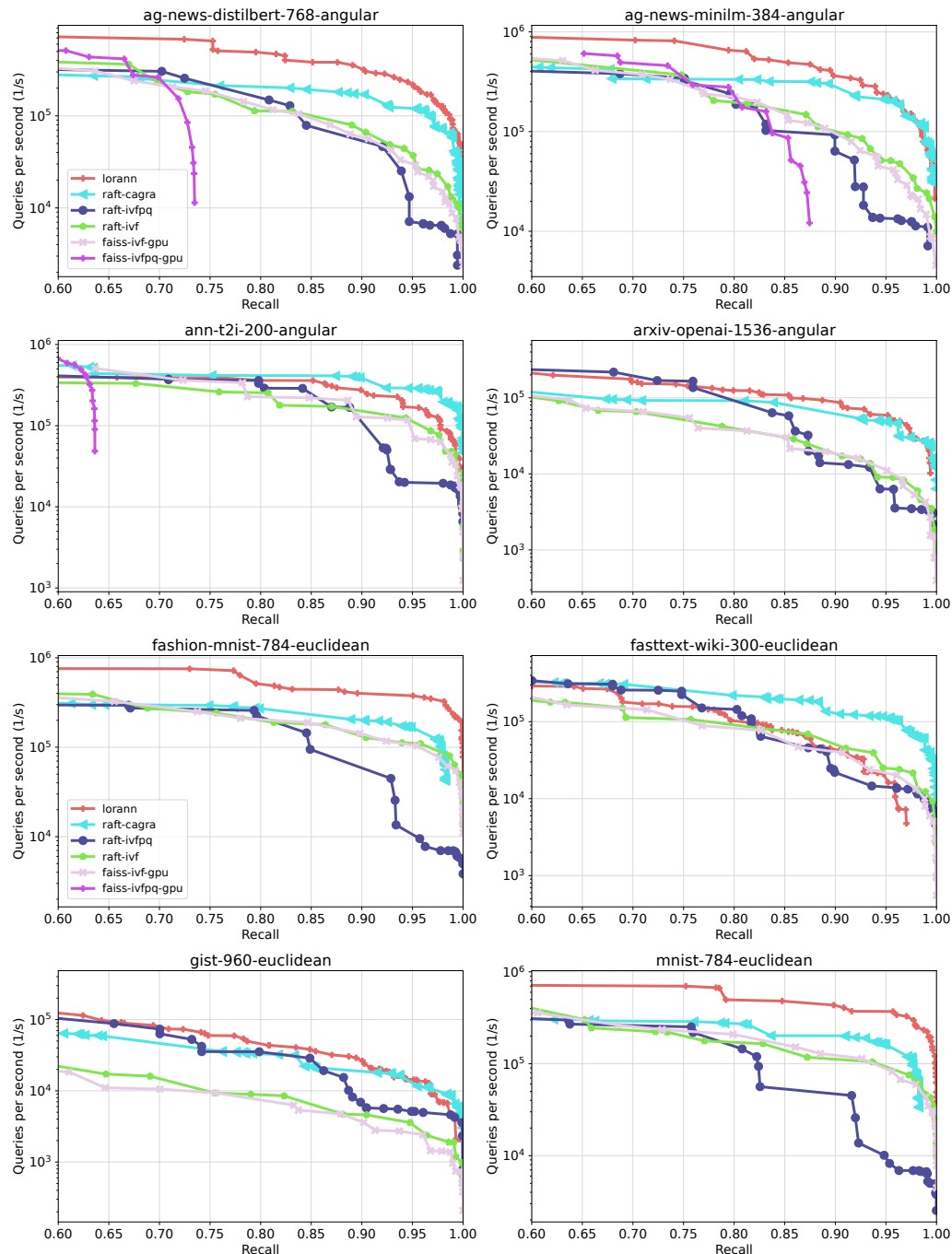

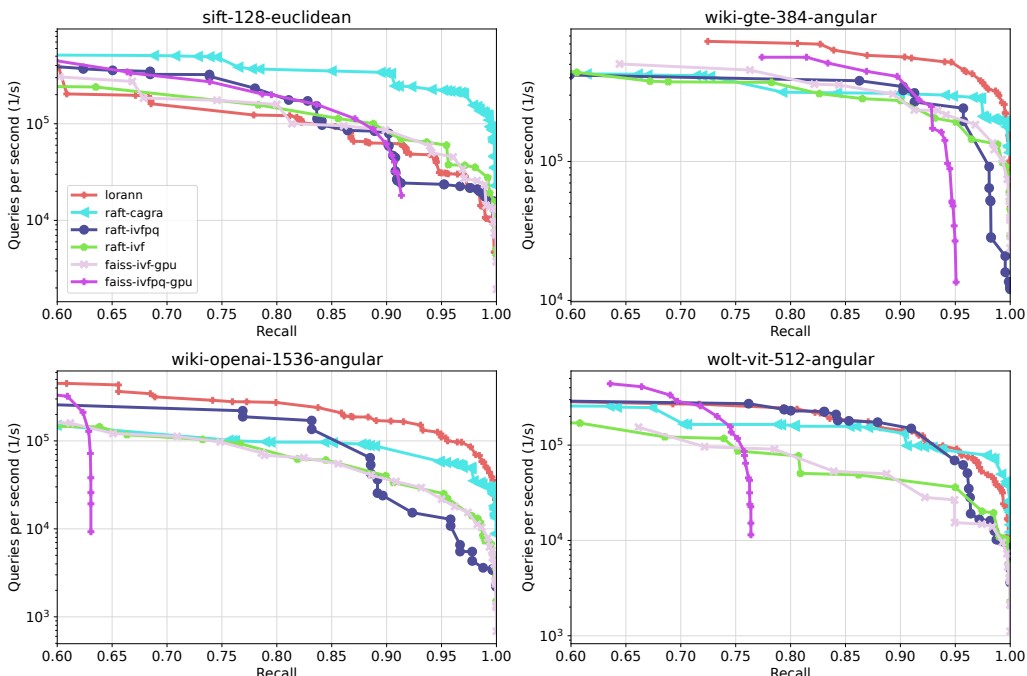

**Apple silicon** In this section, we use MLX[8] to implement `LoRANN` for Apple silicon. We compare MLX versions (both CPU and GPU versions) of `LoRANN` against MLX versions of IVF and the C++ version of `LoRANN` (without quantization) on the Apple M2 Pro SoC. The MLX version of `LoRANN` can take advantage of the M2 Pro GPU and its unified memory architecture to achieve 2–5 times faster query latencies compared to the C++ implementation of `LoRANN`.

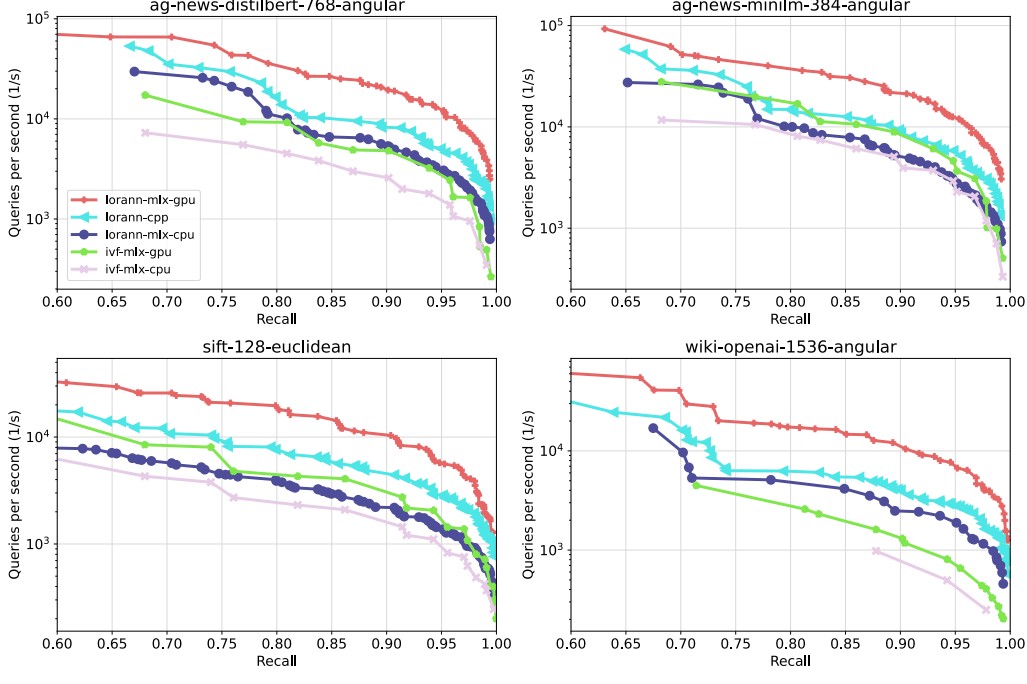

---

[8]`https://github.com/ml-explore/mlx`

