# OpenReview forum: "LoRANN: Low-Rank Matrix Factorization for Approximate Nearest Neighbor Search"
_NeurIPS.cc/2024/Conference — NeurIPS 2024 poster_

### Official Review · Reviewer_udkL · 2024-06-27

**Soundness:** 3
**Presentation:** 2
**Contribution:** 3
**Rating:** 7
**Confidence:** 5

**Summary:**

This paper investigates approximate nearest neighbor (ANN) search, where, given a collection $\mathcal{X}$ of points in $\mathbb{R}^d$, the task is to find the top $k$ data points that are closest to a query point $q$ according to some similarity or dissimilarity measure (denoted by $\delta(\cdot, \cdot)$), such as inner product. There are many classes of algorithms in existence[1], with this particular work falling into the clustering-based paradigm.

In clustering-based (aka Inverted File or IVF) ANN search, $\mathcal{X}$ is partitioned into a set number of clusters, often using a geometric clustering algorithm such as (spherical) KMeans, with every cluster represented using some sketch of the cluster such as its mean. When presented with $q$, the algorithm first identifies $\texttt{nprobe}$ clusters to search by ranking the clusters according to the distance between $q$ and their representative points. It then computes $\delta(q, \cdot)$ with points within the $\texttt{nprobe}$ clusters, and returns the top $k$ points from that set.

This work concerns the second step. Typically the computation of $\delta(q, \cdot)$ uses Product Quantization (PQ) to reduce memory consumption and to perform the distance computation efficiently. Instead, this work reduces the dimensionality of the data matrix within each cluster using its low-rank approximation. The key insight is that, the low-rank approximation is constrained to the space of rank $r$ matrices that predict the inner products well on a specific query distribution.


[1] "Foundations of Vector Retrieval" by S. Bruch. Springer.

**Strengths:**

* The proposed method relies on a very simple yet effective method for supervised dimensionality reduction in the context of ANN search
* The paper is easy to read and arguments straightforward to follow
* Results are encouraging

**Weaknesses:**

Post-discussion Update: The authors have addressed my concerns around the experimental setup, and have expressed interest in adopting a more clear narrative and framing of their contributions.

-----------------

* Presentation:
  - I think the authors can shed quite a bit of fluff by positioning the work as I did in my summary. This work's contribution is very much in the speeding up and improving the accuracy of the score computation phase in clustering-based/IVF ANN search. Presented that way, the authors can immediately focus on the regression problem instead, and not introduce distractions such as the details of clustering, the importance of MIPS (section 2.1), and more. It'd make for a cleaner presentation of your idea, and lets your readers understand the scope of your contributions more clearly.
  - As a minor point, Theorem 1 is a vacuous statement. It's neither necessary to explain the findings of the paper, nor is it insightful enough to birth new research directions. Perhaps you can move it entirely to the appendix if you insist on including it in the work.
  - It must be noted that the method presented in this work is supervised. That is a critical differentiating factor between LoRANN and existing methods such as PQ and Scann.

* Methodology: One of the interesting insights that led to Scann is that not all inner products are equally important. For a quantization method to be successful, it needs to preserve the inner product between $q$ and high-ranking data points better than inner product between $q$ and low-ranking points. In your work, you model the problem as regression, and attempt to minimize the error of inner production approximation equally for all data points. What motivates this uniform weighting? Have you considered a ranking formulation of the problem rather than regression? There is a vast literature on learning-to-rank which, in fact, is very relevant to your idea, but where Scann's insight is baked into its machinery/objective.

* Experiments: Because the methodology is very straightforward and the novelty is minimal, I expect a much stronger experimental evaluation of the proposed method. Here are a few points to consider:
  - Your main experiments conflate two orthogonal axes of evaluation: effect of clustering vs effect of score computation; this I believe stems from the way you present your work. Your contribution, as I noted above, is to the score computation phase of IVF-based ANN search. To evaluate your contributions fairly against SOTA IVF methods, you must partition the data once. Given this fixed set of partitions, you can directly compare the efficacy of LoRANN against PQ and Scann's quantization protocol. By running each method independently as you do now, such that each produces its own partitioning of the data separately, you run the risk of conflating the effect of clustering on IVF's accuracy with the effect of the specific choice of dimensionality reduction/quantization. As it stands, I cannot deduce the exact reason why your method should work better.
  - You are also comparing a supervised method that adapts to a query distribution, with unsupervised baselines. Not only is it not a fair comparison, your results are also not informative. It is not surprising that your method does well: you give it an unfair advantage (as confirmed by Figure 1 - left) by finding a matrix that can predict inner products on *a specific query distribution*. A more reasonable experiment would be to (a) compare a variant of LoRANN that's trained on the data points only (i.e., without training queries) with other IVF methods, and (b) incorporating the query distribution into Scann (its objective can use information about the query distribution). There are other methods that can use a query set to improve quantization ([2,3] are a couple of examples).
  - As a very simple baseline, consider partitioning the data using centroids obtained from a partitioning of queries!
  - Frankly, a comparison with graph methods is nice, but is rather tangential. I encourage you to contrast your method with other IVF methods first, focus your discussion to justifying your proposal against SOTA IVF methods, and then conclude your work with a comparison with graph methods for completeness.


[2] "Query-Aware Quantization for Maximum Inner Product Search" by Zhang et al. AAAI 2023.

[3] "A Learning-to-Rank Formulation of Clustering-Based Approximate Nearest Neighbor Search" by Vecchiato et al. SIGIR 2024.

**Questions:**

My questions mainly concern your experimental evaluation:
* Setup: What's the training set used to train LoRANN? You have kindly given statistics about each dataset, but sadly did not include any information about the size of the query sets.
* Figure 1: What is the size of the initial candidate set in the right figure, where reranking is enabled? It is important to know this because the size of the initial set can explain the small difference between the different curves (e.g., if you retrieve a very large set followed by re-ranking, the accuracy of each method pre-reranking becomes less and less important)
* PQ is obviously sensitive to the bitrate, a hyper-parameter. Can you elaborate how LoRANN holds up against PQ as you sweep the rank parameter and PQ's code size, in terms of speed and memory usage?

**Limitations:**

I really like this work and the simplicity of the idea. I'd love to see this work in print, but I think it can be so much more complete with a proper set of experiments. As it stands, the incomplete experiments limit the reach of this work. A stronger formulation (using ranking, e.g.) can even enhance the results too. I hope my feedback proves helpful in strengthening the arguments of this work.

---

> ### Author Rebuttal · Authors · 2024-08-06
>
> We thank the Reviewer for constructive feedback and good suggestions for improving the manuscript. In particular, clarifying the main contribution of our article as improving the accuracy of score computation in clustering-based ANN search would indeed make the presentation cleaner and our argument stronger; we will revise the manuscript accordingly.
>
> We address the remarks and questions about our experimental methodology by clarifying the details of our experimental setup and providing additional experimental results. Most importantly, we want to clarify that our method does not get an unfair advantage by using any training data the other methods do not have access to. In all the experiments (except the experiments of Section 5 that specifically verify that our method can adapt to the query distribution in the OOD setting), our method uses only the corpus itself as a global training set and draws local training sets as detailed in Section 3. All the data sets, except Yandex used in Section 5, are benchmark data sets for which no specific training set drawn from the query distribution is available. We follow Aumüller et al. [1] by dividing the original data set randomly into a corpus and a set of 1000 test queries. Thus the corpus and the test queries are drawn from the same distribution, and LoRANN exhibits superior performance compared to PQ even in this standard in-distribution setting. This critical detail was missing in the original manuscript, and we thank the Reviewer for pointing this out.
>
> As the Reviewer suggests, it would be natural to also consider non-uniform weightings. The anisotropic quantization used in SCANN works well for GloVE which has a specific data distribution, but does not seems to improve the performance of standard PQ for most data sets as evidenced by both our experiments and experiments by the Faiss authors [2]. We also performed preliminary experiments with optimizing weighted MSE and ranking losses, but they did not improve accuracy while making the models much slower to train. Meanwhile, the simple MSE already results in superior performance compared to earlier methods. However, as the Reviewer suggests, exploring different supervised formulations and loss functions is still an interesting direction for future work.
>
> To address the remark about conflating the effects of clustering and score computation, in Figure 1 of the attached pdf, we perform an additional experiment where the clustering is kept constant to directly compare the proposed score computation method to the score computation method (product quantization) employed by IVF-PQ. The proposed score computation method outperforms product quantization on all of the data sets. Unfortunately, due to the limited time for author response, we were unable to include anisotropic quantization of SCANN in this experiment, but since SCANN did not outperform IVF-PQ in our end-to-end experiments (see Figure 3 and Appendix E of the original manuscript), we do not expect that the performance of anisotropic quantization would be better than the performance of PQ.
>
> To compare the proposed method to PQ for fixed memory consumption, in Figure 2 of the attached pdf, we vary the code size of PQ, and compare LoRANN to IVF-PQ with hyperparameters resulting in similar memory usage. The results of this experiment verify that for fixed memory consumption, LoRANN has superior performance compared to PQ that is a typical choice in memory-limited use cases.
>
> [1] M. Aumüller et al. ANN-Benchmarks: A benchmarking tool for approximate nearest neighbor algorithms. Information Systems 87 (2020): 101374.
>
> [2] Indexing 1M vectors. Faiss Wiki on GitHub.

---

> > ### Comment · Reviewer_udkL · 2024-08-07
> > **Response to Author Rebuttal**
> >
> > Thank you so much for carefully answering all my questions and clarifying your experimental setup! I also appreciate the empirical data you collected for the experiments I requested in my review, in such a short amount of time. This apples-to-apples comparison highlights the strengths of your method even more.
> >
> > I have no further questions or concerns. You have adequately addressed the issues I raised. It is delightful to see such a simple algorithm perform so well in practice.
> >
> > I hope you incorporate my suggestions regarding the structure of the presentation, the reframing of your contributions, and the experiments you've additionally run into a revision of your work.

---

### Official Review · Reviewer_bJwE · 2024-07-15

**Soundness:** 4
**Presentation:** 4
**Contribution:** 3
**Rating:** 5
**Confidence:** 3

**Summary:**

This paper introduces a new method for the nearest neighbor search problem. Leveraging the low-rank assumption, the authors combine low-rank matrix factorization, clustering, and quantization to enhance the speed of nearest neighbor search. The authors conducted extensive experiments to demonstrate the advantages of their method over numerous baselines.

**Strengths:**

1. The authors conducted extensive experiments to compare their methods with other baselines.
2. The method proposed by the authors is easy to follow and implement.

**Weaknesses:**

1. It seems that all the techniques mentioned in this paper have already known to be useful for nearest neighbor search.
2. As shown in Figure 2, all the components contribute to the final results. I don't see any reason why any component applied there is unique to the new algorithm. For example, the clustering and 8-bit quantization techniques appear to be applicable to any existing nearest neighbor search algorithm or library. Thus, I question whether it is fair to employ too many techniques when comparing with other standard nearest neighbor search libraries.

**Questions:**

1. Regarding the low-rank approximation, I don't understand why this method is fundamentally different from first performing dimension reduction on the dataset and then applying any standard nearest neighbor search algorithm.

**Limitations:**

yes

---

> ### Author Rebuttal · Authors · 2024-08-06
>
> We thank the Reviewer for their feedback. However, we want to clarify that our method does not reduce to techniques that have already been used in the earlier ANN literature. As nicely summarized by Reviewer udkL, the main novel contribution of the manuscript is a new supervised method (reduced-rank regression) for cluster-specific score computation in clustering-based ANN search. To the best of our knowledge, this score computation method has not been considered in the earlier ANN literature.
>
> We also want to clarify that the proposed method is fundamentally different from first performing a dimensionality reduction and then applying any ANN algorithm. Our cluster-specific supervised method does not reduce to global dimensionality reduction, since it is neither global nor unsupervised. The key differences are: (1) we compute the scores locally at the clusters by cluster-specific reduced-rank regression models, not by one global model, allowing much greater compression; (2) the proposed score computation method is supervised, i.e., we predict the inner products via reduced-rank regression, and not by an unsupervised dimensionality reduction method.
>
> In addition to the novel score computation method, we propose a concrete ANN algorithm (LoRANN), that combines this score computation method with $k$-means clustering, global dimensionality reduction, and 8-bit quantization. As the Reviewer correctly points out, these techniques have been used earlier in ANN algorithms. However, we combine them in a novel fashion: for example, LoRANN performs the entire score computation using 8-bit integer vector-matrix multiplications. We also do not think that using these techniques makes the comparison to the standard nearest neighbor libraries unfair. We use them to design a practically useful ANN algorithm and to ensure a fair end-to-end comparison with the state-of-the-art ANN libraries that also utilize these techniques.
>
> However, we acknowledge that, as also pointed out by Reviewer udkL, our experimental validation was unclear, since we did not directly compare the novel component of our algorithm (supervised score computation) against the product quantization that is used for score computation by the state-of-the-art clustering-based libraries. In Figure 1 of the attached pdf file, we present the results of an experiment that demonstrates the performance improvement provided by our novel score computation method. In particular, when the clustering is constant across methods, the proposed score computation method outperforms product quantization.

---

> > ### Comment · Reviewer_bJwE · 2024-08-12
> >
> > Thank you for your reply. I have increased my score to 5.

---

### Official Review · Reviewer_rz2s · 2024-07-15

**Soundness:** 4
**Presentation:** 4
**Contribution:** 3
**Rating:** 8
**Confidence:** 1

**Summary:**

The paper describes a method for computing approximate nearest neighbors in
high dimensions. Computing nearest neighbors is a classical problem in
computational geometry, with applications in many areas of computer science.
The classical solutions in low dimensions do not generalize to high dimensions.
The approach in the paper has two main ideas: the first is performing k-means
clustering, computing nearest neighbors on the means, and then computing
more accurate nearest neighbors inside the cluster.
The second is reducing the computation in each cluster to multivariate
regression which can be solved approximately by low rank matrix factorization.

**Strengths:**

The result appears to be very useful in many applications.

**Weaknesses:**

Unfortunately I am not an expert in this field and cannot comment on how this
result compares to the current state of the art.

**Questions:**

N/A

---

> ### Author Rebuttal · Authors · 2024-08-06
>
> We acknowledge that it is not easy  to review an article without field-specific knowledge and appreciate the effort. The ANN-benchmarks project (the link is provided on page 14 in Appendix B of the original manuscript) is the de facto standard for performance evaluation in the field of ANN search. We perform the experiments in the ANN-benchmarks framework, and use their leaderboard to pick the baseline methods. Thus, it should be possible to verify that the comparisons presented in the article are to the actual state-of-the-art even without being an expert in the field.

---

### Official Review · Reviewer_wBA6 · 2024-07-28

**Soundness:** 3
**Presentation:** 2
**Contribution:** 2
**Rating:** 6
**Confidence:** 3

**Summary:**

The paper presents LoRANN, a novel algorithm for Approximate Nearest Neighbor (ANN) search that leverages low-rank matrix factorization and k-means clustering. The core idea is to approximate the ordinary least squares solution of the inner product computation via reduced-rank regression. The authors also introduce a quantized 8-bit version of LoRANN, which is memory efficient and performs well on high-dimensional data. The experiments demonstrate that LoRANN outperforms existing methods on both CPU and GPU.

**Strengths:**

The authors provide extensive experimental results, reporting that their method outperforms leading product quantization-based algorithms and has faster query times than graph-based methods at certain recall levels.

**Weaknesses:**

There exists room for improvement in the visual presentation in this paper. Additionally, it is best to keep the starting or ending points consistent to better compare all methods.

 At different recall levels, LoRANN is sometimes faster, and sometimes slower compared to other methods (GLASS, CAGRA). The authors should analyze the reasons that lead to this phenomenon

**Questions:**

Does LoRANN provide any theoretical guarantees on approximation quality or search time?

**Limitations:**

The limitations of this work have been discussed in the paper.

---

> ### Author Rebuttal · Authors · 2024-08-06
>
> We thank the reviewer for the suggestion on improving the presentation of our graphs and will incorporate this change. We would also be happy to hear any additional suggestions regarding the visual presentation.
>
> As mentioned in Section 8 (Limitations), the reason for the lower relative performance of LoRANN compared to graph algorithms (Glass, CAGRA) at the highest recall levels ($>0.9$) is that our method is clustering-based: too many clusters have to be explored to reach the highest recall levels. This performance degradation at the highest recall levels applies also to other SOTA clustering-based methods such as SCANN and IVF-PQ. Our method improves the performance of cluster-specific score computation, but it cannot overcome this limitation. Naturally, addressing this drawback is subject to further research.
>
> We do not currently provide theoretical guarantees on approximation quality, but this is an interesting direction for future research. The search time of our algorithm can easily be expressed in terms of given hyperparameters, and we will include this detail in a further revision.

---

> > ### Comment · Reviewer_wBA6 · 2024-08-13
> >
> > Thanks for your rebuttal and I will remain my rating.

---

### Author Rebuttal · Authors · 2024-08-06

We thank the reviewers for their constructive feedback. Here we address the most important concerns about novelty (Reviewer bJwE) and experimental methodology (Reviewer udkL) by clarifying our contribution and experimental setup, and performing new experiments:

- We clarify that our method does not get an unfair advantage by using training data that the baseline methods do not have access to. In all of the experiments (except those in Section 5: Out-of-distribution queries) our algorithm draws its training sets from the corpus itself.
- We clarify that the main methodological contribution of the manuscript is a novel supervised score computation method for clustering-based ANN search.
- We empirically verify that the proposed score computation method outperforms the score computation method (product quantization) used by the SOTA clustering-based algorithm IVF-PQ.

Detailed comments are provided in our reviewer-specific responses.

Additionally, the attached pdf includes two new experiments: (1) In Figure 1, we keep the clustering constant to directly compare the proposed score computation method to product quantization (PQ); (2) In Figure 2, we vary the code size of PQ to compare LoRANN to IVF-PQ with different hyperparameters resulting in similar memory usage.

---

### Decision · Program_Chairs · 2024-09-25

**Decision:**

Accept (poster)

**Comment:**

The authors present LoRANN, an innovative method for approximate nearest neighbor (ANN) search that ingeniously combines low-rank matrix factorization, clustering, and quantization techniques. The method is straightforward to understand and implement, and the experimental results are highly promising.

In response to the reviewers' concerns, the authors provide clarifications and additional experiments in their rebuttal, convincingly demonstrating LoRANN's superior performance.

This work offers a simple yet powerful perspective to the field, significantly contributing to its overall value. To further enhance the paper, the authors should focus on refining the presentation in the camera-ready version by more clearly articulating their ideas and the scope of their contributions.

By incorporating these improvements, this paper will make a substantial and valuable contribution to the field of approximate nearest neighbor search.